

# Differential analysis of prokaryotic communities from pristine mangrove tidal zone sediments reveal distinct structures and functional profiles

Carolina Oliveira de Santana[1]; Pieter Spealman[2]; Vânia Maria Maciel Melo[3]; David Gresham[2]; Taíse Bomfim de Jesus[4],

Fabio Alexandre Chinalia[1]

[1]Programa de Pós-Graduação em Geoquímica: Petróleo e Meio Ambiente. Instituto de Geociências (IGEO), Universidade Federal da Bahia (UFBA), R. Barão de Jeremoabo, s/n - Ondina, Salvador, BA 40170-290, Brazil

[2]Center for Genomics and Systems Biology, Department of Biology, New York University, New York, New York, United States of America.

[3]Laboratório de Ecologia Microbiana e Biotecnologia (LEMBiotech), Departamento de Biologia, Centro de Ciências, Universidade Federal do Ceará, Campus do Pici, Bloco 909, Avenida Mister Hull s/n, 60.455-970 Fortaleza, CE, Brazil

[4]Programa de Pós-Graduação em Modelagem em Ciências da Terra e do Ambiente (PPGM), Universidade Estadual de Feira de Santana (UEFS), Avenida Transnordestina, s/n - Novo Horizonte, Feira de Santana, BA 44036-900, Brazil

*Correspondence to*: David Gresham (dgresham@nyu.edu)

# Abstract

Mangrove forests are intertidal ecosystems that constitute a large portion of the world's coastline, as such, they are composed of, and reliant upon, microhabitats defined by the tides. However, we are only beginning to understand tidal



microhabitat biodiversity and their role in nutrient cycling. The majority of metagenomic studies have so far been conducted
on anthropogenically impacted areas. As even mild disruption can severely alter ecosystems and lead to decreased biodiversity
and local extinctions, this is a critical issue . Here, we characterize prokaryotic populations and their involvement in nutrient
cycling across the tidal zones of a pristine mangrove forest within a Brazilian Environmental Protection Area of the Atlantic
Forest. We hypothesize that tidal zones in pristine mangroves constitute distinct microhabitats, are composed of different
prokaryotic communities and, consequently, distinct functional profiles. Samples were collected in triplicate from zones below,
between, and above the tidal waterline. Using 16S rRNA amplicon sequencing, we find significantly different prokaryotic
communities with diverse nutrient cycling related functions, as well specific taxa with varying contribution to functional
abundances between zones. Our findings contrast those observed in anthropogenically impacted mangroves and suggest that
some aspects of mangrove zonation may be compromised by human activity.
**Keywords:** functional prokaryotic ecology; mangrove; metagenomics; tidal zones; prokaryote microbiome; pristine
mangrove forest

# 1. Introduction

Soils are among the greatest sources of microbial diversity on the planet (Tveit *et al.* 2013, Kaur *et al.* 2015, Nesme
*et al.* 2016). These microorganisms are fundamental to many processes such as carbon and nitrogen cycling, as they  shape
and define important characteristics of their habitats through metabolic activities (Wendt-Potthoff *et al.* 2012; De Mandal,
Chatterjee and Kumar 2017; Kumar and Sai 2015). Mangrove ecosystems constitute a large portion of the tropical and
subtropical coastlines of Earth (Yunus *et al.* 2011; dos Santos *et al.* 2011). Beyond their value as natural barriers that reduce
erosion and the impact of storms, they are economically valuable for medicinal, energetic, and eco-tourist uses (Purahong *et*
*al.* 2019), as well as being critical ecosystems in climate change mitigation (Howard *et al.* 2017; Carugati *et al.* 2018). Many
studies have assessed the association between microbial communities from soils and plant development (Panke-Buisse, Lee



and Kao-Kniffin 2017; Wolińska *et al.* 2017; Wagner *et al.* 2014, Zarraonaindia *et al.* 2015, Capdeville *et al.* 2018) with
multiple lines of evidence supporting a plant-soil feedback loop of microbiomes affecting plant diversity while also being
shaped themselves (Van Der Heijden, Bardgett and Van Straalen 2008; Mariotte *et al.* 2018; Bennett and Klironomos 2019;
Miller, Perron and Collins 2019). Thus, considering the dependency of the mangrove forests on the sediment microbiome, it
is important to understand the microbial activities in these sediments in greater detail (Yunus *et al.* 2011; Lin *et al.* 2019).
However, a mangrove forest does not have only a single type of sediment, as tidal ecosystems, they are characterized
by periodic tidal flooding. This dynamic leads to varying environmental conditions across small spatiotemporal scales, with
levels of nutrients, oxygen and salinity periodically fluctuating, resulting in frequent anaerobic conditions and a wide range of
redox potentials (Andreote *et al.* 2012; Lin *et al.* 2019). Dynamic conditions like these can lead to high microbial diversity,
and these microbes play essential roles in the functioning and maintenance of the greater ecosystem (Andreote *et al.* 2012;
Imchen *et al.* 2017; Lin *et al.* 2019; Huergo *et al.* 2018). Although previous research has sought to characterize the prokaryotic
microbiomes across mangrove tidal zones, these works were conducted in anthropogenically impacted areas (Rocha *et al.*
2016; Zhang *et al.* 2018), which, given the sensitivity of the mangrove microbiome, can confound the interpretation of the
community structure (Pupin and Nahas 2014; Alongi 2008; Carugati *et al.* 2018; Nogueira *et al.* 2015).
The Atlantic Forest in Brazil is one of the most biodiverse ecosystems on the planet, containing numerous varieties
of dry and wet broadleaf forests, savannas and mangrove forests, the later of which are primarily composed of genera
*Rhizophora*, *Avicennia*, *Laguncularia* and *Conocarpus* (Pupin and Nahas 2014). This biome is threatened by anthropogenic
disturbances such as logging and farming, as well as habitat loss and fragmentation due to human encroachment, resulting in
a severe decline in its original area (Ditt *et al.* 2013; Ministério do Desenvolvimento Agrário 2010; Pupin and Nahas 2014;
Ghizelini, Mendonça-Hagler and Macrae 2012; Nogueira *et al.* 2015). However, in the southern part of Bahia State, Brazil, a





significant fragment of the Atlantic Forest remains preserved within the Environmental Protection Area (APA) of Pratigi
(Ministério do Meio Ambiente 2004). Recent studies on the environmental conditions of the area show that preservation efforts
initiated in 1998 have been generally effective, resulting in high environmental quality relative to most mangroves, both in
Brazil and globally (Ditt *et al.* 2013; Lopes 2011; Mascarenhas *et al.* 2019). This preserved area constitutes an important site
for the understanding of the ecology of unimpacted mangrove forests.
Therefore, in order to improve our understanding of mangrove ecology, in this study we characterize the prokaryotic
microbiota present in pristine mangrove sediments of the Serinhaém estuary within the Pratigi APA using 16S rRNA amplicon
metagenomics. This approach allows us to identify diverse taxa without the laborious task of culturing them (Kaur *et al.* 2015;
Mocali and Benedetti 2010; Bornemann *et al.* 2015; Nesme *et al.* 2016). Furthermore, we assess the community structure and
functional aspects of these prokaryotes to achieve a deeper understanding of the terrestrial processes at work in different
environments (Mahmoudi *et al.* 2015). Although mangroves have previously been shown to have prokaryotic populations
distinct from the regions they border (ie. mountain forest and restinga), (Mendes and Tsai 2018), only recently has there been
work to understand the differences between mangrove microhabitats (Rocha *et al.* 2016; Zhang *et al.* 2018). Considering
mangrove zonation as driven, primarily, by tide variation, we hypothesized that sediments of different mangrove regions would
differ significantly in richness and composition of prokaryotic communities, with the intertidal zone having the highest
diversity. We assessed the prokaryotic communities, the influence of environmental variables and the functional profile of
these sediments. We also identified the possible taxa driving the different nutrient cycles between zones. Our study provides
insight into the role of microbes in the functioning of mangrove forests and establishes a baseline for monitoring the health of
this important ecosystem.



Importantly, this work was conducted before a massive oil spill occurred off the coastline of Brazil in August 2019,
impacting hundreds of miles of coastline including the Serinhaém estuary where this research was conducted. This work
therefore serves as a baseline measure of the prokaryotic communities of the tidal zones of what was a pristine mangrove
forest. We hope that this will spur subsequent research into the effects that anthropogenic effects have on mangrove
ecosystems.

# 2. Materials and Methods

## 2.1 Study area

The Serinhaém Estuary is located in the Low South Region of Bahia State, Brazil (Fig. 1), between the coordinates
13°35'S and 14°10'S and 39°40'W and 38°50'W. The estuary is within the Pratigi Environmental Protection Area (APA), one
of the few remaining Atlantic forest regions with a total area of 85 686 ha, enclosing a 32 km long portion of the lower Juliana
River and emptying directly into Camamu Bay along with several smaller rivers (Corrêa-Gomes *et al.* 2005).



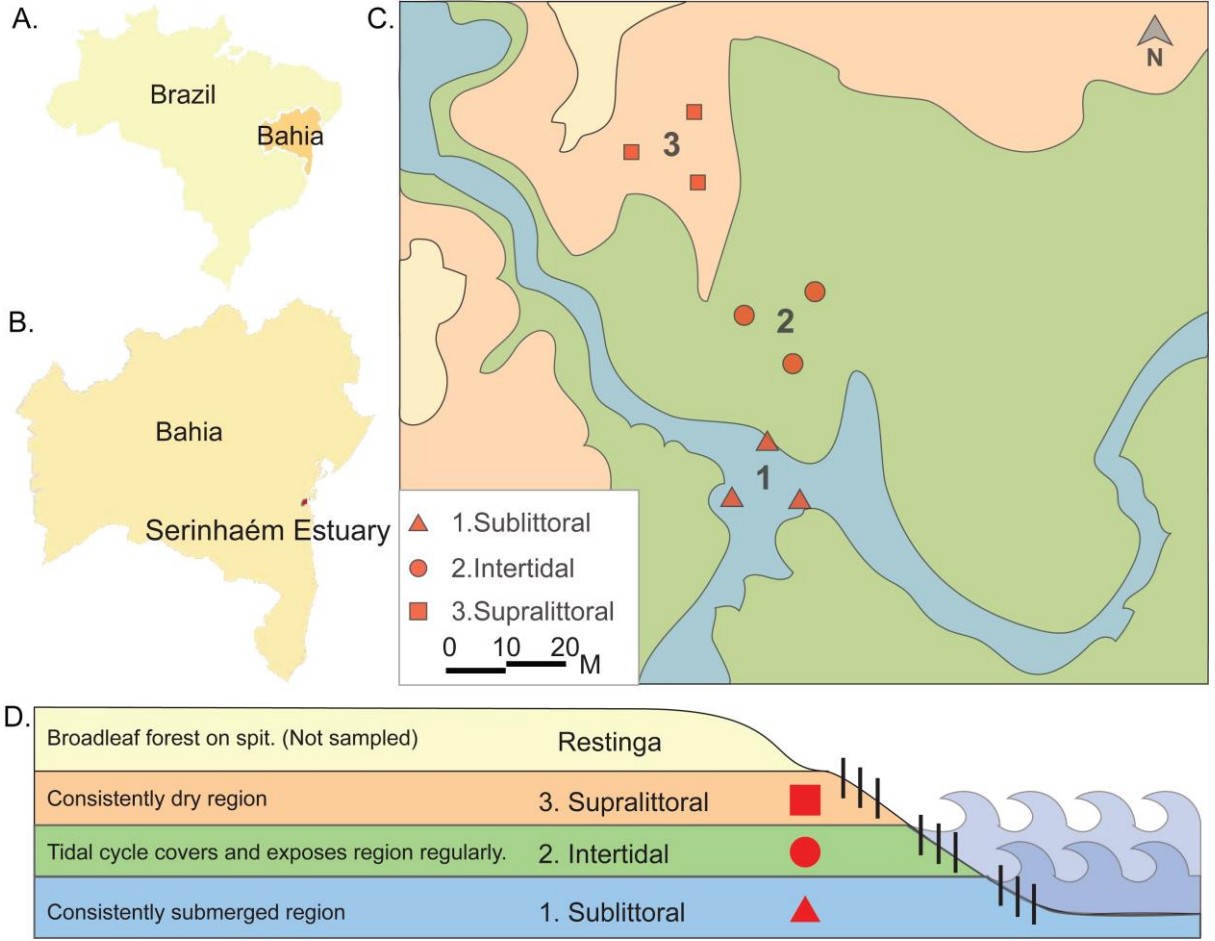

**Fig 1. Map and schematic of sediment sampling sites.** Here we show the locations of the sampling sites relative to Brazil (A) and Bahia (B). A satellite picture shows the relation of the three sampling sites within each zone (1. sublittoral, 2. intertidal, 3. supralittoral, (C)). A schematic shows the topographic and tidal relation of each sampling site (D).

## 2.2 Sampling and DNA extraction

For clarity, we refer to the location of a sample as a 'site' and the collection of sites within a tidal zone as a 'zone'. Samples were collected from 3 tidal zones (centered around 13°42'59.0"S, 39°01'35.9"W) in the Serinhaém estuary in July 2018 during the morning low tide period. No sites exhibited signs of anthropogenic disturbance or pollution. The 3 collection zones were chosen based on tidal influence; sublittoral, intertidal, and supralittoral regions (Fig. 1). From each tidal zone, 3



samples of superficial sediments (top 10 cm of the surface layer) were collected with a cylindrical sediment core sampler. To
ensure that our replicates sampled a broad representation of each zone, sample sites were located a minimum of 15m from
each other in a triangle. Plant and other organic material was manually removed from core samples, with precautions taken to
avoid the disruption of rhizospheres associated with vegetation.
Physical-chemical parameters such as temperature, salinity and dissolved oxygen in the water column were measured
using a multiparameter monitoring system (YSI model 85, Columbus). Each zone had different vegetation densities, with the
sublittoral zone having the greatest plant density, and the supralittoral the least, with almost no vegetation. Metal concentrations
were not collected as previous analysis performed by our lab (Jesus, T.B.) found no significant difference in metal
concentrations relative to background within the Serinhaém estuary. After collection, samples were transferred to the
laboratory. For each sediment core an aliquot was separated and kept in the -20°C freezer for subsequent DNA extraction
while the remainder of the sample was used for measuring organic matter content. The total genomic DNA was extracted from
0.25 g of sediment using the PowerSoil DNA Isolation Kit (Qiagen, Carlsbad, CA, USA). All DNA samples were stored at -
20° C before library preparation and sequencing .

## 2.3 Library preparation and sequencing

After DNA extraction, we used PCR to amplify the V4 region of the bacterial 16S rRNA using the primer pair 515F-
Y (Parada, Needham and Fuhrman 2016) and 806R-XT (Caporaso *et al.* 2011). PCR was performed with a thermal cycler
using the following program: 2.5 µl of each sample were added with 5 µl the forward and reverse primers and 12.5 µl of the
2x KAPA HiFi HotStart ReadyMix, making up for total 25µl and subjected to one cycle of 95°C for 3 minutes, 25 cycles of
95°C for 30 seconds, 55°C for 30 seconds and 72°C for 30 seconds and one cycle of 72°C for 5 minutes. The samples were
amplified in triplicates that were subsequently pooled back as one sample prior to sequencing. After amplification of the V4
region, Illumina sequencing adapters and dual-index barcodes were added to the amplicon target using the Nextera XT indices
Kit according to manufacturer's directions (Illumina, San Diego, CA, USA).The amplified DNA was then checked for size
using a Bioanalyzer. DNA sequencing was performed using Illumina MiSeq platform, V2 kit (300 cycles).



## 2.4 Data analysis

### 2.4.1 Sequence Trimming

Trimmomatic (Bolger, Lohse and Usadel 2014) was used to filter and trim demultiplexed sequences
(ILLUMINACLIP:TruSeq3-PE.fa:2:30:10 LEADING:3 TRAILING:3 SLIDINGWINDOW:4:15 MINLEN:100). A minimum
average read quality score of 15 was required for inclusion while the sliding window cuts any read at the point where the
median quality score over a 4 nucleotide window is less than 15.

### 2.4.2 Sequence denoising and OTU clustering using QIIME2

QIIME (Caporaso *et al.* 2010) was used to join forward and reverse reads into single reads (join_paired_ends.py, -j 4
-p 1). Reads were denoised using DADA2 (Callahan *et al.* 2016) (denoise-single, --p-trim-left 3, --p-trunc-len 0, --p-max-ee
2.0, --p-trunc-q 2) in QIIME2 (Bolyen *et al.* 2019), (q2cli, version 2019.4.0). Denoised sequences are clustered into Operational
Taxonomic Units (OTUs). Alpha-rarefaction was calculated using QIIME2 (alpha-rarefaction, max depth=17000), (S3 Fig.).
We performed a variety of alpha-diversity (S4 Fig., S5 Fig., S6 Fig., S7 Fig.) and beta-diversity (Fig. 3, S8 Fig.) tests using
QIIME2 (core-metrics-phylogenetic, p-sampling depth 9340).

### 2.4.3 Taxonomic assignment, community visualization and environmental tests

Taxonomic assignment used Vsearch (Rognes *et al.* 2016) in QIIME2 using Open Reference with 97% similarity (--
p-perc-identity 0.97) against the reference 16S rRNA sequences in SILVA database (Silva SSU 132), (McDonald *et al.* 2012).
QIIME2 visualizations for 1. OTU abundance (S1 Fig., S2 File), 2. proportional representation between sites, are available as
a supplementary files (S2 File), and 3. taxonomy (S3 File).
Phylogenetic reconstruction was carried out in QIIME2 using the representative sequences for each OTU and a
QIIME2 feature classifier trained using the 97% similarity representative set containing only 16S rRNA sequences (e.g.
silva_132_97_16S.fna). All groups were required to be present within at least 2 samples with a minimum of 3 reads each.



QIIME2 tree files were accessed in R using QIIME2R (version 0.99.12). Tree visualization (Fig. 7) was performed
with R (version 3.4.4) using Metacoder (Foster, Sharpton and Grünwald) (version 0.3.2). Posterior analysis was performed
using Phyloseq (McMurdie and Holmes 2013), (version 1.22.3). Analyses in R were plotted using ggplot2 (McMurdie and
Holmes 2013; Villanueva and Chen 2019).
Vegan (Dixon 2003), (version 2.5-6) was used to test correlations between community structure and environmental
variables. Distances were calculated using metaMDS, (engine=monoMDS, try=1000, k=3), and then fit the environmental
variables using envfit (default settings, permutations=333), (S4 Table).

## 2.4.4          Functional          analysis          using          PICRUSt2

Functional analysis was performed using PICRUSt2 (version 2.3.0-b) (Douglas *et al.* 2019; Barbera *et al.* 2019;
Czech, Barbera and Stamatakis 2020; Louca and Doebeli 2018; Ye and Doak 2009) with default settings. Both the Kegg
Orthologs (KOs) and MetaCyc pathways were analyzed for significant (p-value <= 0.05) differential abundances after centered
log-ratio transformation (aldex.clr) using the general-linear model method (aldex.kw) of the ALDEx2 package (ver 1.18.0). A
heatmap of KOs with differential abundance between sample sites was then generated (Fig. 6).

## 2.4.5 Site-specific taxonomic and functional enrichment

In order to identify which species were significantly different in abundance in each zone we performed taxa
enrichment analysis (Fig. 5), (Spealman *et al.* 2020). First, OTU abundances were normalized by downsampling to match the
least abundant zone (Intertidal). Taxa abundances are the sum of all assigned OTU abundances. For each taxa, we required
that a significant difference be found between sites using a Chi-squared, 2x3 test, with correction
(scipy.stats.chi2_contingency) using the mean normalized abundance. To correct for false positives due to variance between
replicates we required the distributions of unnormalized OTU abundances between sites to also be significantly different
(Mann-Whitney U test, scipy.stats.mannwhitneyu, p-val <= 0.05). Finally, to ensure biological relevance, we required the
effect size to represent at least 5% difference in log-fold abundance between sites.



To determine which taxa were associated with differences in functional abundance, we also calculated KO enrichment
specific to each taxa at a given taxonomic level (Fig. 5, 7) (Spealman *et al.* 2020) using the functional abundance results of
PICRUSt2. We required that a given taxa must have at least 10% of all KO functional abundance at the given level; that the
functional abundance be significantly enriched using a Binomial exact test (Bonferroni corrected p-value <= 0.05), and the
taxa must have at least three distinct KOs within a single pathway that meet these criteria. All KOs and their metabolic
pathways are available in a supplemental file (S6 File).

## 2.4.6 Accessibility

The entire computational workflow is available on Github: https://github.com/pspealman/COSantana_2020.
Data used in the performance of the analysis and archival versions of the computational workflow are available on Dryad:
https://doi.org/10.5061/dryad.gf1vhhmkz    (Spealman    *et*    *al.*    2020).    [    Temporary    link    for    reviewers:
https://datadryad.org/stash/share/bwmAgXaOhXT2JNHKbfX15wpIJ3dAxhOXrnjdwnwSSHM ]
The    data    has    been    deposited    as    PRJNA608697    in    the    NCBI    BioProject    database:
https://www.ncbi.nlm.nih.gov/bioproject/PRJNA608697

# 3. Results

## 3.1 Taxonomic composition of prokaryotic communities

After quality filtering, a total of 204 599 bacterial and archaeal sequences remained for community analysis,
corresponding to an average of 22 733.2 sequences per sample. Sequence clustering yielded a total of 1709 OTUs. Of these,
1,623 OTUs and 193 143 sequences were assigned to Bacteria (94.4%) and 84 OTUs and 10 707 sequences were assigned to
Archaea (5.2%) kingdoms (S1 Fig.). 749 sequences clustered in 2 OTUs (0.4%) that could not be assigned to any prokaryotic
kingdom. Additionally, one mis-annotated Archea taxa originally named "uncultured eukaryote" has been manually changed





to "SUE" for SILVA uncultured eukaryote. All sites combined, we identified 37 unique phyla, 142 classes, 165 families, 142
genera and 97 species. From the total, 18 087 sequences (approximately 9%) weren't assigned to the phylum level. More than
88% of all the sequences that could be assigned to the phylum level belonged to 6 phyla: *Proteobacteria* (30.3% abundance,
62 135 sequences), *Firmicutes* (29.4% abundance, 60 307 sequences), *Chloroflexi* (6.4% abundance, 13 225 sequences),
*Planctomycetes* (5.3% abundance, 10 888 sequences), *Actinobacteria* (4.6% abundance, 9390 sequences) and *Crenarchaeota*
(3.8% abundance, 7 921 sequences). The total sequence and OTU abundances in all the observed phyla are summarized in S1
Table. Fig. 2 shows all the classes and orders of the 6 dominant phyla in the data set. Families and genus are shown in
Supplemental (S2 Fig.).

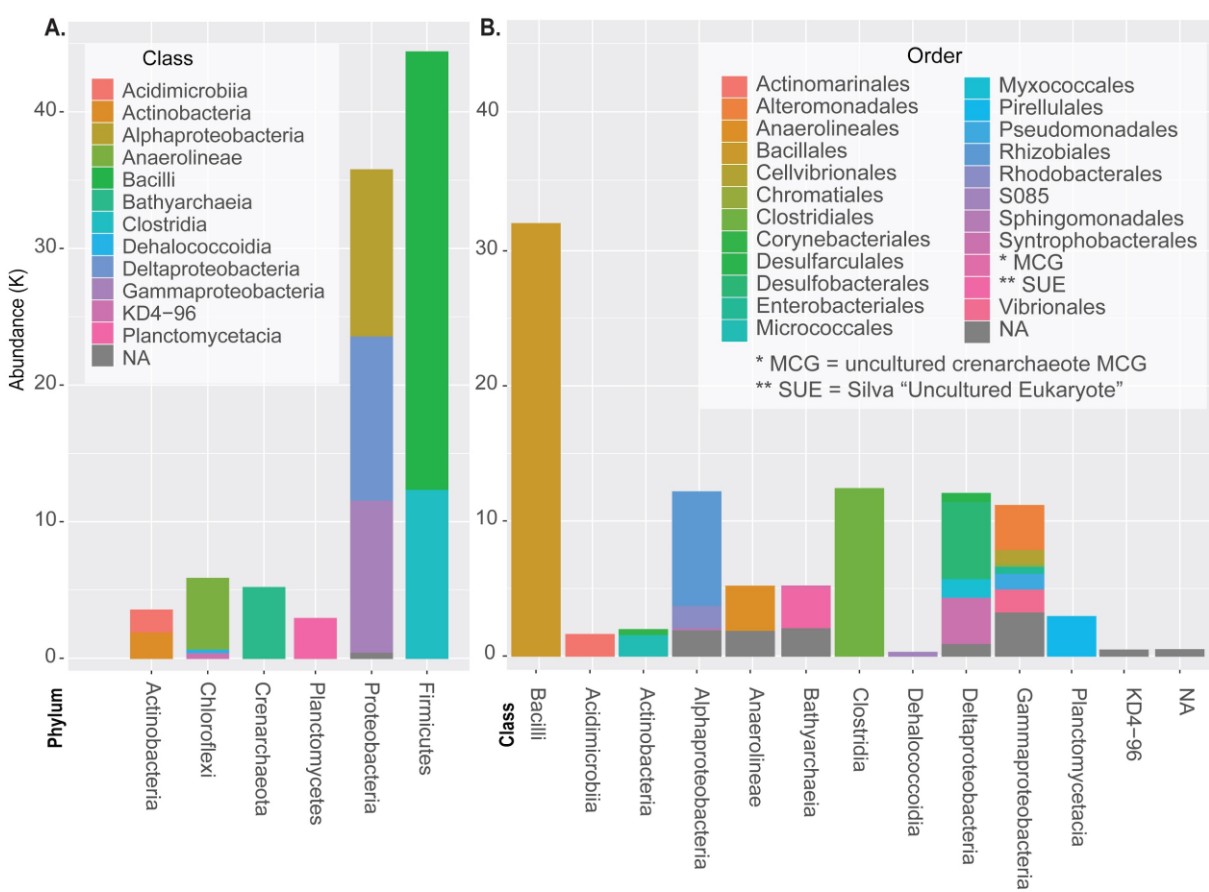




**Fig 2. Taxonomic abundances from all sample sites.** Taxa identified within the samples are shown as stacked bar plots, the horizontal axis
is the higher taxonomic level while the stacked bars are the lower level. Phylum is the horizontal axis with Class being the stack (**A**), Class
is the horizontal axis with Order being the stack (**B**).

## 3.2 Microbial diversity of mangrove tidal zones

The alpha diversity indices for each zone were calculated using QIIME2 (S4 Fig.). Overall, the sediments from the
sublittoral zone had higher richness and diversity indices, while the intertidal zone exhibited the lowest alpha diversity indices
(S5, S6, S7 Fig.).
We used the beta-diversity package of QIIME2 to assess differences in prokaryotic populations between zones (Fig.
3), (S8 Fig.) and significant differences using the distance metrics in S2 Table. Using the Bray-Curtis and Jaccard distance
metrics (Fig. 3) we found significant differences in population structures between zones (p-value < 0.05 in Jaccard, p-value <
0.1 in Bray-Curtis, $\alpha$= 0.1, 90% confidence). The pseudo-F test results for pairwise PERMANOVA failed to indicate statistical
differences site to site, despite the apparent dissimilarity between the groups in the plots, potentially because this test would
require a larger number of samples (see Supplemental Permanova Section).

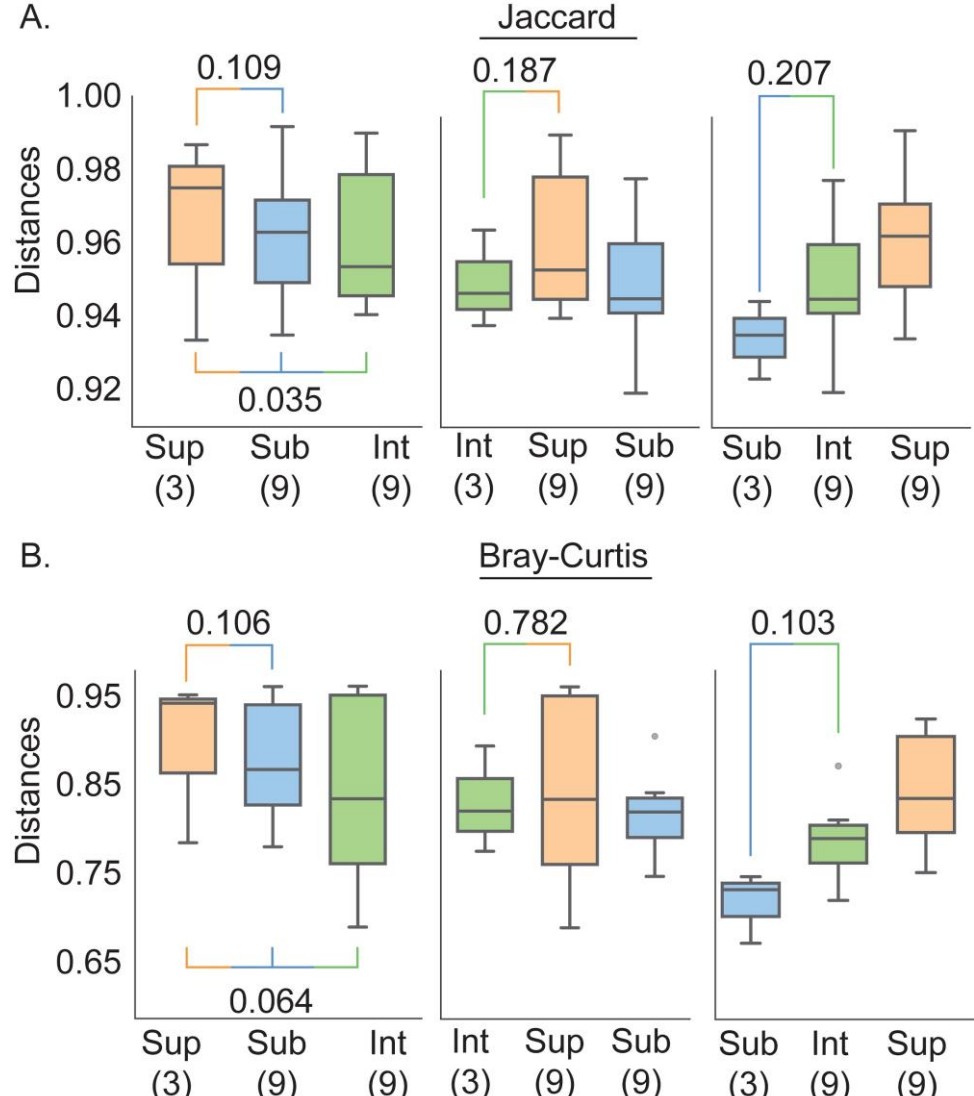


**Fig 3**. **Beta-diversity between sampling sites**. All sites are compared pairwise using distance metrics. The differences between sites are

significant for both the Bray-Curtis and Jaccard distance metrics ($\alpha = 0.1$).

## 3.3 The influence of environmental variables

To determine if differences in population structures between zones correlated with abiotic environmental variables

we measured salinity, water content, organic matter and temperature from each tidal zone (S3 Table) and associated the





taxonomic abundances from each sample from each zone using Vegan (Dixon 2003) (see Methods). The results revealed a
significant correlation between the prokaryotic populations within each zone and salinity and organic matter (Fig. 4, S4 Table).
Specifically, increased organic matter was positively correlated with sublittoral population abundances, while increased
salinity was positively correlated with supralittoral population abundances. Neither water content or temperature measures
reflect a significant difference in community structure between zones.

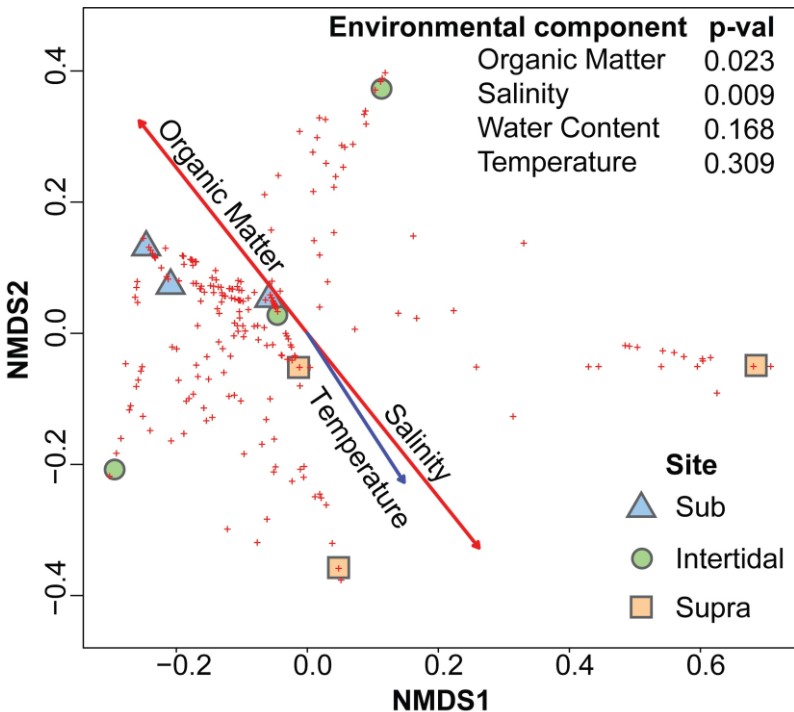


**Fig. 4. Correlation between environmental variables and prokaryotic communities.** Dots represent taxonomic abundances per site as
plotted by Vegan metaMDS using nonmetric multidimensional scaling (NMDS). Arrow length is a representative of the predictor strength
of environmental variable vectors, with red arrows having statistical significance as calculated by envfit (p-value <= 0.05).

## 3.4 Taxa enrichment by tidal zones of the mangrove

To determine what specific taxa were significantly different in abundance between zones, we performed a zone-
specific enrichment test for each taxa, (Fig. 5). Briefly, this method tests the normalized abundances of each taxa to identify





taxa with statistically significant and large effect size differences in functional abundances between zones (see Methods
section). Nearly every taxa (96%, 25/26) identified by this method has its greatest abundance in the sublittoral zone, with 38%
(10/26) having an inverse relationship between elevation and abundance, such that the abundance increases from the
supralittoral to intertidal to sublittoral zone.

233       We used PICRUSt2 (Douglas *et al.* 2019) to correlate identified taxa with Kegg Orthologs (KOs) and then calculate

the functional abundance of KOs, (S4 File), (see Methods). Families with enrichment of metabolism associated KOs,
(accounting for more than 10% of the total of a given KO) for at least 3 KOs of a single metabolic pathway were then labelled
with an icon for that pathway (Fig 5.). (see Methods).

237       We found that 7 of the 8 enriched families make substantial (>10%) contributions to carbon metabolism associated

KOs (specifically methane metabolism), with the exception of *Anaerolineaceae*. Similarly, 6 families contribute for numerous
sulfur metabolism associated KOs such as assimilatory sulfate reduction contributed by *Bacillaceae*, and dissimilatory sulfate
metabolism associated KOs contributed by *Syntrophaceae, Desulfobulbaceae,* and *Desulfobacteraceae*. The families
*Bacillaceae, Desulfobulbaceae,* and *Desulfobacteraceae*, produce substantial amounts of nitrogen metabolism associated KOs,
with *Bacillaceae* contributing to dissimilatory nitrate reduction to ammonium (DNRA) and *Desulfobulbaceae* contributing to
nitrogen fixation associated KOs. Only *Anaerolineaceae* and *Bacillaceae* contribute to phosphorus metabolism associated
KOs. Taken together, this suggests that zone-specific taxa enrichment may also contribute to differential metabolic activities
at these zones. We describe the potential metabolic roles of each of these and other families below.







**Fig. 5. Site specific measures of taxa find significant enrichment in sediments of different tidal zones.** Here we show taxa with

significantly different abundances between zones (Chi2, p-val <= 0.05), statistically different distributions of abundance (MWU, p-val

<=0.05), and whose mean effect size exceeded 10% (see Methods).



## 3.5 Site specific differences in metabolism associated Kegg Orthologs

To determine if there were significant differences in metabolic activity between tidal zones we calculated the functional abundance of metabolic KOs for each zone (see Methods). We found 22 metabolism associated KOs with significantly different functional abundance between zones (Fig. 6). 8 KOs show higher abundances in the sublittoral sediments, above both intertidal and supralittoral, including both members of the phosphorus D-galacturonate degradation pathway. Conversely, only one KO is enriched in the supralittoral sample above both sublittoral and intertidal zones, and no KO is enriched in the intertidal zone. 5 KOs exhibit an abundance in negative proportion to elevation, such that sublittoral is greater than intertidal, which is, in turn, greater than supralittoral. Only one KO is observed to have the opposite trend, with highest abundance in the supralittoral sites and lowest in sublittoral. Finally, 7 KOs show a bimodal distribution with near equal abundances between sublittoral and supralittoral, with intertidal being the significantly less, including all four of the nitrogen metabolism associated KOs. Taken together, KO enrichment reinforces the previously observed trend of reduced abundance in the Intertidal site, and greatest abundance at the Sublittoral zone.





* accd6 = acetyl-CoA/propionyl-CoA carboxylase carboxyl transferase subunit
** eda = 2-dehydro-3-deoxyphosphogluconate aldolase / (4S)-4-hydroxy-2-oxoglutarate aldolase
*** nxrA = nitrate reductase / nitrite oxidoreductase, alpha subunit
**** nxrB = nitrate reductase / nitrite oxidoreductase, beta subunit

263

**Fig. 6. Zone specific measures of metabolism associated Kegg Orthologs.** The heatmap shows the 22 metabolism associated KOs that have significant functional abundance differences between zones for Carbon, Phosphorus, Sulfur and Nitrogen pathways.



## 3.6 Nutrient cycling pathways and taxa identified within communities

In order to identify potential functional roles of the prokaryotic communities, we used PICRUSt2, which associates KEGG orthologs (KOs) with 16S rRNA amplicons through gene family associations to reference genomes (Douglas *et al.* 2019), (S4 File shows KO assignment and abundance for each taxa). Here, we show the functional profiles for carbon, nitrogen, phosphorus, and sulfur metabolic pathways. Where possible we rely on previous work that has identified microbial groups with specific metabolic activity as potential drivers for processes such as methanogenesis, nitrogen fixation and nitrification (Levipan *et al.* 2016; Fierer 2017). This is complemented with measured functional abundances observed for KOs associated with the carbon, nitrogen, phosphorus, and sulfur metabolic pathways across the three mangrove intertidal zones. We use this information to link taxa found in this study to their most probable and relevant nutrient cycling activities (Fig. 7, a full version with all taxa is available as a supplemental file S5 File).

### 3.6.1 Carbon cycling

Carbon cycle pathways such as methane oxidation and methanogenesis showed enrichment in the sublittoral zone. We find that *Syntrophaceae* contributes significantly with different methanogenesis associated KOs including pyruvate ferredoxin oxidoreductase subunits alpha, beta, delta, and gamma (15%, 24%, 43%, and 14%, respectively) and heterodisulfide reductase subunits A2, B2, and C2 (30%, 27% and 36%, respectively), (S4 File). We also find plentiful *Archaeal* families contributing the majority (>50%) of methanogenesis KOs; *Nitrosopumilaceae*, uncultured families of *Lokiarchaeia*, and *Bathyarchaeia* (see S4 File).

### 3.6.2 Sulfur cycling

Significantly higher abundances of sulfur transformation KOs were found in the sublittoral zone. The family *Rhodobacteraceae* (Delmont *et al.* 2015) contributes substantial abundances (>10%) of 12 different sulfur metabolism KOs



(S4 File). The families *Syntrophaceae*, *Desulfobacteraceae*, *Desulfobulbaceae* contribute to almost 90% of the KOs associated
with dissimilatory sulfate reduction, which is in accordance with previous observations (Kuever 2014, Wörner and Pester
2019, Wiegel, Tanner and Rainey 2006; Oren and Xu 2014; Meyer *et al.* 2016).
Members of the order *Rhizobiales* are observed to be major drivers of the sulfur oxidizing process, as they contribute
85% of the sulfur-oxidizing protein SoxY and 65% of SoxZ. The family *Chromatiaceae,* which makes up the majority of
purple sulfur bacteria, are known for their role in the sulfur cycle in numerous environments (Wiegel, Tanner and Rainey 2006;
Oren and Xu 2014; Meyer *et al.* 2016; Hanada and Pierson 2006; Xia *et al.* 2019) and contribute substantial amounts of sulfur-
oxidizing protein SoxA (18%), SoxB (24%), SoxX (18%), and SoxZ (17%).
As a side note, *Desulfatiglans*, of the family *Desulfarculaceae* are also reported to act in degrading aromatic
hydrocarbons (Sun *et al.* 2010; Jochum *et al.* 2018). We find that they contribute 26% of all the 3-oxoadipate enol-lactonase,
and nearly 99% of all benzoyl-CoA reductase subunit BamB, BamC, and various benzoate degradation associated enzymes.

## 3.6.3 Phosphorus cycling

Similar to what is observed for the other nutrients, P cycling KOs are, overall, more abundant in the sublittoral
sediments. The genus *Pseudomonas* that belongs to the family *Pseudomonadaceae*, contributes a substantial amount (>40%)
of three phosphorus metabolism associated KOs and nearly 99% of three others. Along with *Pseudomonas,* the genus *Bacillus*
from the family *Bacillaceae*, has been suggested to be active in P cycling in mangrove sediments through phosphate
solubilization (Kalayu 2019; Malboobi *et al.* 2009). In our analysis we found that *Bacillus* contributed to phosphorus
metabolism associated KOs at the genus level, with 18 KOs greater than 20%, 5 of which are greater than 50%.

## 3.6.4 Nitrogen cycling





We observed significantly lower abundances for nitrogen transformation pathways for the intertidal zone. In the
literature, members of the family *Pirellulaceae* are relevant for ammonia oxidation processes (Jiang *et al.* 2015). In our
analysis, *Pirellulaceae* contributes only minorly to nitrogen metabolism with only 7 KOs having greater than a 10%
contribution. However, the ammonia-oxidizing archaea families represented by *Nitrososphaeraceae* (Kerou *et al.* 2016) and
*Nitrosopumilaceae*, make up nearly the entirety of the nitrification associated methane/ammonia monooxygenase KOs subunits
A, B, C (72%, 72%, 56% and 28%, 28%, 44%, respectively). Several components of the dissimilatory nitrate reduction to
ammonium (DNRA) pathway showed significantly different abundances between zones due to variations in the genus *Bacillus*.
*Desulfobulbaceae* members are also major contributors to nitrogen fixation, specifically nitrogenase iron protein associated
KOs.
Members of the *Clostridiaceae,* one of the most abundant families in the samples, are known for participating in
nitrogen-fixing, as well as other nitrogen transformations (Wiegel, Tanner and Rainey 2006; Oren and Xu 2014; Meyer *et al.*
2016; Chen, Toth and Kasap 2001). We found the genus *Clostridium* sp. AN-AS6E to significantly contribute to nitrogen
metabolism KOs. The genus *Vibrio* contributes greater than 20% to 3 nitrogen metabolism associated KOs, the genus
*Marinobacter* with 5 KOs, and the genus *Bacillus* with 10 KOs.
Other families that have known capacity for nitrogen fixing in the sediments are *Flavobacteriaceae* (Kämpfer *et al.*
2015), represented by 7 genera, *Pseudomonadaceae* (Özen and Ussery 2012), *Spirochaetaceae* (Lilburn *et al.* 2001), and
*Rhizobiaceae* (Broughton 2003), although only *Flavobacteriaceae* makes significant contributions to nitrogen metabolism
KOs. Some members of *Chromatiaceae* are known to be active in the nitrification process, as the genus *Nitrosococcus*
(Campbell *et al.* 2011) as well as the family *Pirellulaceae* (Kellogg, Goldsmith and Gray 2017), although we do not find these
making significant contributions in terms of KOs. Finally, organisms capable of performing ammonification are represented



by *Micrococcaceae* (Dastager *et al.* 2014) and *Rhodobacteraceae* (Delmont *et al.* 2015), although only *Rhodobacteraceae* has
significant contributions; with 4 nitrogen metabolism associated KOs.











**Fig. 7. Phylogenetic tree showing additional metabolic data.** Phylogenetic tree depicting assigned node abundances (Nodes, represented
by color and thickness of branches) and whether a given taxa in associated with a metabolic pathway given either literature (dashed red line)
or significant enrichment of functional abundance of metabolism associated KOs (see Methods). Names are only shown for nodes with
associated literature citations or leaves with significant enrichment. A more complete tree displaying the names of all taxa is available as a
supplement (S5 File).

# 4. Discussion

Previous work has shown that mangrove forests have variation in community structure, often associated with different
biotic and abiotic factors, however, the majority of these have been conducted in anthropogenically impacted areas (Pupin and
Nahas 2014; Marcial Gomes *et al.* 2008; Alzubaidy *et al.* 2016; Rocha *et al.* 2016; Ceccon *et al.* 2019; El-Tarabily 2002;
Imchen *et al.* 2017; Lin *et al.* 2019; Zhang *et al.* 2018), confounding the makeup of the microbial populations, their abundance,
and determination of environmental influences on these population structures. Importantly, the majority of this work does not
consider or does not identify the mangroves under study as anthropogenically impacted, despite frequently being only a few
km from dense metropolitan and industrial centers (Imchen *et al.* 2017; Lin *et al.* 2019; Zhang *et al.* 2018; Ceccon *et al.* 2019).
Notably, studies that sought to identify differences induced by pollution and urbanization on mangroves did find large-scale
differences in prokaryotic populations in impacted areas compared to preserved mangrove areas (Pupin and Nahas 2014;
Nogueira *et al.* 2015). While pioneering, this research did not study the population differences of distinct microhabitats within
mangroves. Here, we extend the study of preserved mangrove areas to characterize prokaryotic populations within tidal zone
microhabitats.
Previous work that has focused on mangrove microbial diversity has found that composition of bacterial communities
in sediments correlates with a broad range of variables, such as; hydrodynamic regimes, granulometry, organic matter content
(Colares and Melo 2013), vegetation (Rocha *et al.* 2016) and pollutant distributions, all of which can generate niche variations
(Peixoto *et al.* 2011). However, ecosystems can also exhibit a robust community structure, such that even significant
differences in variables, such as pH, are mitigated, resulting in less variation between communities than expected (Huergo *et*
*al.* 2018). Previous work by (Gong *et al.* 2019) found environmental conditions and historical events play an important role in





shaping the bacterial communities as well. In our study, we found both salinity and organic matter to be significantly correlated
with community populations in different tidal zones (Fig. 4). While mangrove degradation has long been known to be sensitive
to both of these environmental variables (Alongi 2015) these results diverge from others (Rocha *et al.* 2016; Zhang *et al.*
2017), making it difficult to infer general trends. While this work adds to our understanding of prokaryotic variation in
mangrove forests, it is important to note that more studies need to be performed in mangroves more diverse than the
anthropogenically impacted areas of South America and Asia, as the majority of them have been so far (Imchen *et al.* 2017;
Lin *et al.* 2019; Huergo *et al.* 2018; Zhang *et al.* 2018; Gong *et al.* 2019; Ghizelini, Mendonça-Hagler and Macrae 2012).
The results of the alpha-diversity tests showed a greater number of OTUs and a greater taxonomic diversity in the
sublittoral mangrove sediments, while the intertidal zone had the lowest richness and diversity. Differentiation in mangrove
sediment communities (as measured by beta-diversity) from zones with distinct biotic and/or abiotic characteristics has
previously been reported in the literature (Alzubaidy *et al.* 2016; Rocha *et al.* 2016; Peixoto *et al.* 2011; Jiang *et al.* 2013;
Ceccon *et al.* 2019).
While differences in environmental variables between sites may partially explain differences in prokaryotic
communities between tidal zones (Fig. 5) it is also possible that they are influenced by additional factors, such as fungal,
eukaryotic microbe, and plant rhizome contamination (Rocha *et al.* 2016; Zhang *et al.* 2017). The presence of microbes
typically associated with plant rhizosphere, has been observed in many previous studies (Alzubaidy *et al.* 2016; Rocha *et al.*
2016; Gomes *et al.* 2010; Zhang *et al.* 2018; Ceccon *et al.* 2019). In these studies, the rhizosphere sediments confer enrichment
of specific taxa, and have higher alpha-diversity relative to non-rhizosphere associated sediments. While we attempted to avoid
the inclusion of plant material in our collection of sediments, the presence of mangrove trees and other vegetation is an
unavoidable feature of the tidal zones. Similarly, the higher density of vegetation observed in the sublittoral area may, in part,
explain the higher diversity of the prokaryotic populations we identified there (Bennett and Klironomos 2019; Miller, Perron
and Collins 2019). Additionally, the microbiome of the mangrove can be heavily influenced by eukaryotic communities
(Alzubaidy *et al.* 2015), which would be invisible to our 16S rRNA amplicon sequencing method. We believe our





understanding of prokaryotic community structures is only one step in a larger process that should ultimately include rhizome,
fungal, and eukaryotic populations information.
Functional profiling is a proxy measurement of metabolic activity, and where possible we attempted to supplement
the taxonomic and functional profiles generated by our analysis with controlled metabolic studies from the literature. The
identification of a rich and divergent set of taxa associated with the diverse nutrient cycles in mangrove sediments was expected
due to the previous observations of the microbial diversity in these environments (Rocha *et al.* 2016; Ceccon *et al.* 2019;
Cabral *et al.* 2016; Mendes and Tsai 2014; Zhao and Bajic 2015) and was both confirmed and extended in this study as we
identified further distinctions between tidal zones. Notably, the majority of metabolism associated KOs (Fig. 5, 6) and
pathways (S9 Fig.) had higher abundance in the sublittoral mangrove sediments. The higher abundances of KOs found at the
sublittoral zone is likely due to the greater taxonomic diversity that was also observed for this region.
Our data suggests that the intertidal regions of mangrove forests have lower prokaryotic diversity than those in the
constant environments in the supra- and sublittoral regions. While this is consistent with some microbial models of
microhabitat diversity (Alongi 1988) it is in disagreement with more recent studies of microbial communities in tidal zones
that suggests higher diversity is maintained by the dynamic tidal environment (Lv *et al.* 2016; Ceccon *et al.* 2019; Imchen *et*
*al.* 2017; Lin *et al.* 2019; El-Tarabily 2002; Imchen *et al.* 2017; Lin *et al.* 2019; Zhang *et al.* 2018). One possible explanation
of this is that the specialization we observe at the constant environments (sub- and supralittoral) would be lost in mangroves
that are degraded, polluted, or otherwise anthropogenically impacted, as these zones would be rapidly colonized by
opportunistic species. We hypothesize, that in pristine mangroves, the constant environments offered by sub and supralittoral
microhabitats allow for the accretion of specialists, while the frequent and cyclic variation of the tides act as a selective
pressures on microbial communities, making the intertidal zone inhospitable for organisms specialized to the supra- and
sublittoral environments. Conceptually, this property of a dynamic environment defining a selective niche, similar to a physical
barrier, is worthy of further study.
As noted previously, the Serinhaém Estuary, where this work was conducted has since been impacted by a large off-
shore oil spill, the effects of which are unknown. It remains an open question if the sublittoral zone's greater abundance of
taxonomic diversity and enrichment in metabolic function correlate with a resilience to environmental perturbations. One could



hypothesize that the combination of diverse communities with organisms possessing redundant metabolic functions may be
more stable against perturbations as the larger standing variation of organisms will respond differently to stressors, thus
increasing the likelihood of the survival of some taxa (Girvan *et al.* 2005). However, while the diversity of taxa at the sublittoral
site may grant it certain advantages, in terms of being a more robust ecosystem, it is also in a more perilous position as the
water itself is frequently the carrier of contamination from rivers, as is the case for urban waste (Yunus *et al.* 2011), and from
the oceans through the tides, as the case with oil spill contamination (Cabral *et al.* 2016). Thus, it is important to consider that
different parts of the mangrove tidal zone would be exposed in different levels of contamination and that this could affect the
organisms in a site-specific manner. We hope that our work in characterizing what was once a pristine mangrove forest aids
the further exploration of the impact anthropological activities have on the microbial communities of mangrove ecosystems.

# 410 5. Funding

This work was supported by the Coordenação de Aperfeiçoamento de Pessoal de Nível Superior – Brasil (CAPES) –
Finance Code 001 (https://www.capes.gov.br/) and by the Fundação de Amparo à Pesquisa do Estado da Bahia – Fapesb
(http://www.fapesb.ba.gov.br/) [grant number PNE0021/2014]. Additional funding provided by National Science Foundation
(NSF) [MCB1818234].

# 416 6. Acknowledgements

The authors would like to thank the sequencing facility of the Microbial Ecology and Biotechnology Laboratory (LEMBiotec).

# 419 7. Conflict of Interest

The authors have declared that no competing interests exist.



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
