# Peer review of "Effects of tidal influence on the structure and function of prokaryotic communities in the sediments of a pristine Brazilian mangrove"

_Biogeosciences, 2020_

## Referee Comment (RC1) · Anonymous Referee #1 · 17 Sep 2020

General comments: This study collected samples from 3 tidal zones of a pristine mangrove habitat for 16S amplicon sequencing and analysis. There needs to be a massive overhaul of the writing and synthesis of the results in order for this to be a publishable piece of work. Generally, there are a lot of writing issues with the current manuscript, with grammar/spelling errors, typos, and run-on sentences throughout. The introduction needs the most work. There are abrupt transitions between paragraphs, and many concepts are not adequately introduced. Specifically, the introduction needs to explain why it is important to study the different tidal zones in a mangrove habitat, why microbes are important to mangroves specifically, and what the broader implications are for this work. The results section is too long and can be shortened. Much of the results describe methods which are already present in the methods section. The results also have discussion and implications that should be left in the discussion section. The PI-CRUSt results section reads like a combined results/discussion section, when it should be just the results. The discussion section primarily covers how current findings corroborate or conflict with previous findings. There is little synthesis of results, though the last two paragraphs of the discussion touch upon the beginnings of what could be synthesized from these results. There also needs to be discussion of the limitations of using PICRUSt. The wording throughout the manuscript needs to be changed to reflect the nature of the data (that this was done using 16S data, not metagenomes). What is the novelty of this study, what are its contributions, and why is it important? These are especially vital questions to answer because there are already studies characterizing pristine mangrove microbiomes. The discussion needs to highlight how the findings fit into large ecological processes happening in the mangrove tidal zones, and not just a rehash of existing literature.

Specific comments: Line 21 mentions past metagenomic studies, but line 27 states this study used 16S rRNA amplicon sequencing. Currently it seems like the authors are using the two terms interchangeably.

Line 29: I think there should be some elaboration of how findings from this study contrast results from anthropogenically impacted mangroves in the abstract. It's not informative to the reader to just state that there is a difference.

Line 40: Explain the role of mangrove ecosystems in climate change mitigation

Line 45: This paragraph concludes with acknowledging the dependency of mangrove forests on the sediment microbiomes, but the first paragraph in the introduction wasn't written in a way that convinces me of this dependency. How, specifically, do sediment microbes benefit mangroves?
Line 51: How does increased microbial diversity lead to an enrichment of microbes that play essential roles in ecosystem functioning? Which specific taxa are enriched by these dynamic conditions that would go on to maintain ecosystem functioning?

Line 54: This is the first time that the sensitivity of the mangrove microbiome is introduced, which I found to be really surprising. I think there should be more of a lead up to this statement (what is the microbiome sensitive to? How is that sensitivity manifested?)

Line 68: I have a problem with the term "16S rRNA amplicon metagenomics". 16S amplicon sequencing and metagenomics are two very different techniques.

Line 76: Why did the authors hypothesize that the intertidal zone would have the highest microbial diversity? And also what would this mean ecologically? These types of information needs to be included in the introduction, particularly since mangrove zonation is not introduced until the last lines of the introduction.

Line 80: But aren't there other studies that have looked at mangrove microbiomes under pristine conditions? I.e. Nogueira et al 2015. I don't have a problem with that, but this introduction is written in a way that implies this is the first study to look at pristine mangrove sediment microbiomes.

Figure 1: Explain what Abundance (K) means in the figure caption

Figure 3: Why present results from both Jaccard and Bray-Curtis beta diversity metrics?

Figure 3: Include descriptions of (A) and (B) in figure caption.

Results throughout: Do not need to include methods or even say "see methods" in the results section. See lines 216 and lines 228 as examples.

Line 234: This sentence is written confusingly and needs further elaboration. Does the word "Families" refer to protein families or microbial families? Additionally, are these

BGD
just metabolism-associated KOs of KOs associated with all pathways?

Line 245: Don't need this last sentence.

Line 261: This last sentence is a far overreach and makes no sense. The data does not support or even show there being a reduced abundance of KOs between the sites/zones.

Line 261: Are site and zone used interchangeably here?

Line 273: Functional abundances for KOs weren't measured...they were extrapolated from the 16S data.

Line 378: These functional profiles are predicted from amplicon data, and were not actually measured in this study.

Technical corrections: Line 25, abstract: Sentence should be rewritten. It's grammatically awkward as is.

Line 37/38: remove the "the" before tropical, and remove "of Earth"

Line 47: Run-on sentence

Line 57: "latter" instead of "later"

Line 64: "relative to most mangrove forests" instead of of "relative to most mangroves"

Line 117: missing "of" in "with 5 ul the forward..."

Line 150: Sentence is grammatically awkward. Rephrase.

Line 188: Avoid contractions

Line 193: Plural use of families and singular use of genus

Line 239: Setence is grammatically awkward. Rephrase.

Figure 5 caption is written awkwardly and should be rephrased.
KEGG should be capitalized throughout.

Figure 6: elemental pathways should not be capitalized.

---

## Referee Comment (RC2) · Anonymous Referee #2 · 20 Oct 2020

General Comments: Clearly a lot of work went into this study, however there is still much work that needs to be done with the paper. Currently the paper reads like a draft that still needs several more rounds of circulation between authors. The introduction and discussion sections need better organization/flow with more specific, relevant background/literature pertaining to sediment microbes and mangroves, and importance of tidal zone (how that might influence microbial communities and mangroves). There are numerous sentences within the results section that belong under either the methods or discussion sections. The discussion lacks focus and synthe-

sis. Overall, this paper needs a significant overhaul. It seems like it would be useful for the authors to clarify specific objectives, if not for the paper, for themselves, to achieve better focus and clarity in conveying this study and its findings. Another major consideration is that the authors should be very careful about what and how they convey findings and conclusions on metabolic pathways and functionality of microbes when only using 16S rRNA amplicon data. Particularly, they should be weary and cautious about using a tool like PICRUST2 to make any major conclusions with respect to microbial functionality. Personally, I think that if you are going to use PICRUST2 as a tool here you should be backing up as much of those findings with literature as possible. For example, you could compare your findings with mangrove metagenomics studies such as those done by Andreote et. al. 2012. Also, it is important to very explicitly state the pitfalls of PICRUST2 analyses (see Sun et. al. 2020, https://link.springer.com/content/pdf/10.1186/s40168-020-00815-y.pdf) and that these are just inferential findings and would need to be confirmed via additional analyses such as transcriptomics or experimental setups.

Specific Comments:

Abstract

Line 21: The term metagenomic at this point is used solely to describe shotgun or whole metagenomic sequencing, not amplicon sequencing.

Line 27: You say "significantly different prokaryotic communities but in Line

Line 31: Change metagenomics to "amplicon" or "16S rRNA". I don't think you should include "function in the keywords, as functionality is solely inferred indirectly via amplicon sequencing and Picrust2 analysis. Choose either "Mangrove" or "Pristine Mangrove Forest".

Introduction

Line 37: Be more specific when you say "large portion", how much do they constitute?

Line 42: Instead of talking about studies which look at microbial communities and plant development, include specific background on microbes and mangroves that is relevant to your study.

Line 45: You haven't really established "dependency" of mangrove forest on sediment microbiome at this point. You can expand on how they can be considered dependent or remove this type of wording.

Line 47: I'm not sure what you mean by "single type of sediment." Since you don't discuss types of sediment or measure sediment characteristics such as grain size or grain type (silt, sand, etc.) in this paper you should not mention sediment type. I think you might mean that due to the fact that mangroves are highly influenced by tidal flow, which results in variations of "environmental conditions across small spatiotemporal scales,"

Line 60: What is the "original area"? Do you have specific information on this?

Line 71: Not sure what you mean by "terrestrial processes." Do you mean biogeochemical processes?

Line 74: I would say something more like "understand the differences between micro-habitats within mangrove systems" instead.

Line 75: By "mangrove regions" do you mean different tidal zones? If not, you might want to briefly explain what the different regions of mangroves are.

Line 78: Use something like "We identified taxa which may be driving different utrient cycles between zones." I should note that you may want to rethink this sentence altogether as you don't really show that there are different nutrient regimes/cycles between different zones as of now. You could include literature that suggests this or data of your own to support it.

Lines 81 -85: You could circle back to this in the discussion, specifically you could theorize what changes in community structure you might expect based on your findings

and the literature in contaminated mangrove sediments.

Methods

Line 100: What is meant by tidal influence? It would be good to include demarcations for this, i.e. distances, vegetation, etc. Based on Fig.1 it seems like you may have used sediment water content, or time of water coverage.

Line 104: When you write "disruption of rhizospheres" do you mean "contamination" of rhizospheres associated with vegetation, because you aren't wanting to include those communities?

Line 107: Were vegetation densities measured, if so, what was the metric?

Line 111: How was organic matter content measured?

Line 127: I don't think you need the "ILLUMINACLIP" section, especially if you already have your code published on Github. Additionally, you explain your code in text immediately following.

Line 131: You mention just QIIME, and QIIME2 in the following steps. QIIME is no longer supported or kept up, so if you used the original QIIME I would recommend using QIIME2 for that step.

Line 132 – 141: DADA2 calls ASVs and not OTUs. I have only superficially used QIIME2, as I typically use mothur or DADA2 directly in R, but my understanding is that QIIME2 and DADA2 primarily call ASVs, but gives the option to then cluster those ASVs after they have been called. If this is what you did, you should explain that process briefly. In reading on (Line 138), it would seem that there is either a miscommunication or misunderstanding of the bioinformatic steps with respect to clustering and taxonomic assignment. "Open reference" refers to a clustering method which can be done using Vsearch in QIIME2. It looks like when assigning taxonomy after OTU clustering, Q2 gives you 3 different option, I am thinking you used classify-consensus-vsearch? I would also combine sections 2.4.2 and

2.4.3 in to one section where all bioinformatics workflows in QIIME 2 are discussed together. For reference: https://docs.qiime2.org/2020.8/tutorials/otu-clustering/ and https://docs.qiime2.org/2020.8/tutorials/overview/

Line 149 – 151: What dissimilarity metric did you use for metaMDS, i.e., jaccard, bray-curtis etc.? I am wondering why you didn't run something like Canonical Analysis of Principle Coordinates (capscale in Vegan) to investigate correlations between environmental variables with community structure.

Line 153: Did you take the limitations in the link provided below into consideration when running these analyses?

PiCRUST2 Limitations Link: http://picrust.github.io/picrust/tutorials/quality_control.html

Line 159: Explicitly explain how you conducted the taxa enrichment analysis. I could not find the reference paper for Spealman et. al. 2020.

Results

Line 186: Do you mean uncultured "prokaryote" not "eukaryote"? I don't believe there is any eukaryotic designations in the 16S SILVA taxonomy reference. You should also qualify further why you felt comfortable assigning an archaeal taxon to the uncultured "eukaryote".

Line 196: Figure 1B colors are difficult to differentiate. Consider adding a pattern to colors which are too close to differentiate. For figure 1B, consider changing the y-axis range to 32 so that we can see more of the other bars.

Line 201, 204, 205: QIIME2 sentences belong in methods

Line 207: P-value for Bray-Curtis is not significant

Line 211: Why did you use both distance metrics, i.e., Jaccard and Bray-Curtis? You should choose the one most appropriate to your data and study. Did you try hierarchical clustering to see how data might cluster without apriori considerations like tidal zone?

Line 222: What is your organic matter (OM) metric, is it total OM? I don't think this is the best/clearest way to analyze these data with environmental variables. See methods comment Line 145-151.

Line 227-229: These sentences belong in the methods section

Line 231: What is "elevation" in this context? Do you mean zonation?

Line 236: What is an "icon" in this context?

Line 237: What specifically about "carbon metabolism"? As all microbes need a carbon source, this should be explained in a bit more detail.

Line 247: In figure 5 it would be good to use different colors for differentiating bacteria and archaea as you are already using green and blue in the figure legend.

Line 252: This sentence belongs in Methods

Lines 255 & 256: don't need the "above both" phrase, it is confusing.

Line 261/262: "Taken together, KO enrichment reinforces the previously observed trend of reduced abundance in the Intertidal site, and greatest abundance at the Sublittoral zone." This sentence seems like it should be in the discussion, especially if the "previous trend" you are referring to is one form the literature. Also, be consistent on whether you are capitalizing the tidal zones or not. I think it is more correct not to capitalize.

Line 268-276: I may have missed the results in this paragraph, but it seems like all of this belongs in the methods section as it is describing how something was done versus reporting the findings of what was done.

Line 270: Where is here? Is it this study or a figure?

Lines 277 – 327: I would combine all of the "cycling" sections under one section called "biogeochemical cycling" or something like this. You have several sentences throughout this section that would be more appropriate in the discussion section. Essentially

any of the sentences with citations should probably be in the discussion. Examples: Lines 287, 291-293, etc.

Line 328: I appreciate the effort that went into this figure. Personally, I would like to see this figure with less taxa, only ones you specifically mention within the text, so that it is less busy. I am not sure why the # of nodes legend needs to be a tapered triangle. It makes me think I should be considering both the color and thickness of lines when I'm looking at nodes. I also think you should use a more differentiating way to denote taxa with an associated metabolic role in the literature and taxa with KO greater than 10%. You could use black and white circles, and add a gray circle for those with both if they exist.

Discussion

I decided to make an overarching comment here, instead of going line by line, because the discussion needs a lot of work and restructuring. I think one of the best ways to go about fixing the discussion will be to come up with a thesis statement for each paragraph and figure out what points you are trying to convey. This will help you to clarify and re-organize your thoughts. I would like to see in the discussion more synthesis of your findings, such as why you think you might find certain taxa with potential metabolic capabilities enriched in certain tidal zones. It seems to me that you set out to study the sediment microbiome of pristine mangrove environment across 3 tidal zones to serve as a baseline and to characterize differences in taxa and potential biogeochemical cycling that is predominant in those zones. However, neither your introduction or discussion provide enough clear, relevant background or support for your overarching goal. I would have also liked to see some discussion on anaerobic taxa, and where you find more anaerobic taxa with respect to tidal zone. You us the term microhabitat in both the intro and discussion, but it is unclear what this means in the context of your study. You should clearly define what your usage of microhabitat means. Are you talking about microbial habitat, are you taking about spatial scales, millimeters – meters?

Technical Corrections:

Line 57: Latter not "later"

Line 62: I am assuming that the A in APA is referring to Ambiente, but just want to point out you use the English "Environmental" just before APA, so I'm not sure if you should write EPA or use the word Ambiente.

Line 117: Use protocol instead of "program"

Line 154: All KEGGs should be capitalized.

Line 193: Genera instead of "genus".

Line 101: I don't know if superficial is the correct word, I typically see the use of "surficial".

---

## Author Comment (AC1) · 10 Nov 2020

**Reviewer comments are in black, our responses are in red.**

**Reviewer #1**

**General comments:**

This study collected samples from 3 tidal zones of a pristine mangrove habitat for 16S amplicon sequencing and analysis. There needs to be a massive overhaul of the writing and synthesis of the results in order for this to be a publishable piece of work.

Generally, there are a lot of writing issues with the cu
rrent manuscript, with grammar/spelling errors, typos, and run-on sentences throughout. The introduction needs the most work. There are abrupt transitions between paragraphs, and many concepts are not adequately introduced.

Specifically, the introduction needs to explain why it is important to study the different tidal zones in a mangrove habitat, why microbes are important to mangroves specifically, and what the broader implications are for this work.
The results section is too long and can be shortened. Much of the results describe methods which are already present in the methods section. The results also have discussion and implications that should be left in the discussion section. The PICRUSt results section reads like a combined results/discussion section, when it should be just the results. The discussion section primarily covers how current findings corroborate or conflict with previous findings. There is little synthesis of results, though the last two paragraphs of the discussion touch upon the beginnings of what could be synthesized from these results.
There also needs to be discussion of the limitations of using PICRUSt.
The wording throughout the manuscript needs to be changed to reflect the nature of the data (that this was done using 16S data, not metagenomes).
What is the novelty of this study, what are its contributions, and why is it important?
These are especially vital questions to answer because there are already studies characterizing pristine mangrove microbiomes.
The discussion needs to highlight how the findings fit into large ecological processes happening in the mangrove tidal zones, and not just a rehash of existing literature.

We appreciate the reviewers comments and thank them for their time and careful reading of our manuscript. We have revised the manuscript to address each of the reviewer's comments. We believe that the revised document more clearly articulates 1) why the study of tidal zones in pristine mangrove areas are important and how this research differs from previous work, 2)  the methods that we have used and their potential limitations, and 3) how this work contributes to a deeper understanding of microbial ecology in mangrove areas. Furthermore, we have added a supplemental table with a short review of contemporary research on prokaryote populations in pristine and impacted mangrove tidal zones to aid the reader and provide additional context for our study. Throughout the manuscript the text has been revised and shortened, with references to methodology removed from the results section and a revised discussion section that

emphasizes the results and how our findings contribute to our understanding of  this important ecology.

**Specific comments:**

Line 21 mentions past metagenomic studies, but line 27 states this study used 16S rRNA amplicon sequencing. Currently it seems like the authors are using the two terms interchangeably. We have changed the terms throughout the text to make clear that our approach is 16S rRNA amplicon sequencing and to distinguish this approach from metagenomics.

Line 29: I think there should be some elaboration of how findings from this study contrast results from anthropogenically impacted mangroves in the abstract. It's not informative to the reader to just state that there is a difference. We have rewritten the abstract and introduction to greater contrast the relation between this work and previous work on mangroves, and specifically mangrove tidal zone microhabitats.

Line 40: Explain the role of mangrove ecosystems in climate change mitigation. We have removed text referring to global climate change for concision.

Line 45: This paragraph concludes with acknowledging the dependency of mangrove forests on the sediment microbiomes, but the first paragraph in the introduction wasn't written in a way that convinces me of this dependency. How, specifically, do sediment microbes benefit mangroves? We have re-written the text to clarify how the sediment microbiome is important for the greater ecosystem functioning.

Line 51: How does increased microbial diversity lead to an enrichment of microbes that play essential roles in ecosystem functioning? Which specific taxa are enriched by these dynamic conditions that would go on to maintain ecosystem functioning?
We have re-written the text to clarify that the statements about microbial roles in ecosystem functioning refer to general features of sediment microbiomes, as has been reported  in the literature.

Line 54: This is the first time that the sensitivity of the mangrove microbiome is introduced, which I found to be really surprising. I think there should be more of a lead up to this statement (what is the microbiome sensitive to? How is that sensitivity manifested?) We have added a section, and supporting literature, that details the sensitivity of mangrove microbiomes to pollution, sea level, salinity change, and environmental degradation.

Line 68: I have a problem with the term "16S rRNA amplicon metagenomics". 16S amplicon sequencing and metagenomics are two very different techniques. The text has been corrected

and now we  refer to our method as 16S rRNA amplicon sequencing here and throughout the manuscript.

Line 76: Why did the authors hypothesize that the intertidal zone would have the highest microbial diversity? And also what would this mean ecologically? These types of information needs to be included in the introduction, particularly since mangrove zonation is not introduced until the last lines of the introduction. We have edited the text to introduce the concept of zonation at the beginning of the introduction and make clear that the hypotheses are based on previous ecological studies reported in the literature. We have also added an explanation of one ecological interpretation of this hypothesis and the consequences of our rejection of the hypothesis.

Line 80: But aren't there other studies that have looked at mangrove microbiomes under pristine conditions? I.e. Nogueira et al 2015. I don't have a problem with that, but this introduction is written in a way that implies this is the first study to look at pristine mangrove sediment microbiomes. We apologize for the confusion, we have amended the text and added a supplemental table to more clearly communicate the novelty and context of this work. The reviewer is correct that there are numerous studies of microbes in pristine mangroves. However the majority of these have not focused on microhabitats created by tidal variations, but only accessed the differences in communities between the impacted and pristine mangroves. To the best of our knowledge only two papers have assessed prokaryote population differences between mangrove tidal zones and both of these were performed in anthropogenically impacted mangroves. We have changed the text to clarify this difference between our study and these previous studies.

Figure 1: Explain what Abundance (K) means in the figure caption We have added this information to the figure caption.
Figure 3: Why present results from both Jaccard and Bray-Curtis beta diversity metrics? We now report only one distance metric (Jaccard) for the figure in the main text and the figures for additional tests were moved to the supplemental section.
Figure 3: Include descriptions of (A) and (B) in figure caption. The caption now refers to the A) and B) parts of the figure.
Results throughout: Do not need to include methods or even say "see methods" in the results section. See lines 216 and lines 228 as examples. We have excluded the mentions to methods in the text throughout the results section.

Line 234: This sentence is written confusingly and needs further elaboration. Does the word "Families" refer to protein families or microbial families? Additionally, are these just metabolism-associated KOs of KOs associated with all pathways? We apologize for the confusion.  We have corrected the text and clarified that these are only metabolism-associated KOs and that we refer to taxonomic families.
Line 245: Don't need this last sentence. The sentence has been removed.

Line 261: This last sentence is a far overreach and makes no sense. The data does not support or even show there being a reduced abundance of KOs between the sites/zones. We have removed the sentence from the text.

Line 261: Are site and zone used interchangeably here? We have edited the text so that only the term 'zone' is used to refer to microhabitats.

Line 273: Functional abundances for KOs weren't measured...they were extrapolated from the 16S data. This text has been amended to clarify this distinction.

Line 378: These functional profiles are predicted from amplicon data, and were not actually measured in this study. We have corrected the text to clarify our approach and its inherent limitations.

**Technical corrections:**

Line 25, abstract: Sentence should be rewritten. It's grammatically awkward as is. We have rewritten the sentence.

Line 37/38: remove the "the" before tropical, and remove "of Earth" We have corrected the text.

Line 47: Run-on sentence We have corrected the text.

Line 57: "latter" instead of "later" We have corrected the text.

Line 64: "relative to most mangrove forests" instead of of "relative to most mangroves" We have corrected the text.

Line 117: missing "of" in "with 5 ul the forward..." We have corrected the text.

Line 150: Sentence is grammatically awkward. Rephrase. We have clarified the sentence.

Line 188: Avoid contractions The text has been corrected.

Line 193: Plural use of families and singular use of genus The text has been corrected.

Line 239: Sentence is grammatically awkward. Rephrase. We have corrected the sentence.

Figure 5 caption is written awkwardly and should be rephrased. We have corrected the text.

KEGG should be capitalized throughout. We have corrected the text.

Figure 6: elemental pathways should not be capitalized. We changed the text on the figure.

---

## Author Comment (AC2) · 10 Nov 2020

**Reviewer comments are in black, our responses are in red.**

**Reviewer #2**

**General comments:**

Clearly a lot of work went into this study, however there is still much work that needs to be done with the paper. Currently the paper reads like a draft that still needs several more rounds of circulation between authors. The introduction and discussion sections need better organization/flow with more specific, relevant background/literature pertaining to sediment microbes and mangroves, and importance of tidal zone (how that might influence microbial communities and mangroves). There are numerous sentences within the results section that belong under either the methods or discussion sections. The discussion lacks focus and synthesis. Overall, this paper needs a significant overhaul. It seems like it would be useful for the authors to clarify specific objectives, if not for the paper, for themselves, to achieve better focus and clarity in conveying this study and its findings. Another major consideration is that the authors should be very careful about what and how they convey findings and conclusions on metabolic pathways and functionality of microbes when only using 16S rRNA amplicon data. Particularly, they should be weary and cautious about using a tool like PICRUST2 to make any major conclusions with respect to microbial functionality. Personally, I think that if you are going to use PICRUST2 as a tool here you should be backing up as much of those findings with literature as possible. For example, you could compare your findings with mangrove metagenomics studies such as those done by Andreote et. al. 2012. Also, it is important to very explicitly state the pitfalls of PICRUST2 analyses (see Sun et. al. 2020, https://link.springer.com/content/pdf/10.1186/s40168-020-00815-y.pdf) and that these are just inferential findings and would need to be confirmed via additional analyses such as transcriptomics or experimental setups.

We thank the reviewer for their time and insightful comments. We have significantly revised our manuscript to address their concerns and are confident that the paper has substantially improved as a result. As per their advice, we have re-written the introduction to clarify the intent of this paper and situate it within the larger body of mangrove metagenomic studies. We have also sought to make clear the limitations of our methodologies. We particularly appreciate the reference to the Sun et al. 2020, which was published the same month we had submitted our manuscript. By incorporating the findings of Sun et al we have highlighted the intrinsic and extrinsic limitations of PICRUSt2 in relation to our data.

**Specific Comments:**

**Abstract**

Line 21: The term metagenomic at this point is used solely to describe shotgun or whole metagenomic sequencing, not amplicon sequencing.

We have corrected the text throughout our manuscript to appropriately refer to the method used as 16S rRNA amplicon sequencing.

Line 27: You say "significantly different prokaryotic communities but in Line

This comment appears to be truncated.

Line 31: Change metagenomics to "amplicon" or "16S rRNA". I don't think you should include "function in the keywords, as functionality is solely inferred indirectly via amplicon sequencing and Picrust2 analysis. Choose either "Mangrove" or "Pristine Mangrove Forest".
We agree with the reviewer and have corrected the text.

**Introduction**

Line 37: Be more specific when you say "large portion", how much do they constitute?
We have revised this to more specifically describe their size.

Line 42: Instead of talking about studies which look at microbial communities and plant development, include specific background on microbes and mangroves that is relevant to your study.
We agree with the reviewer that this was an overly broad background and we have now condensed the references to focus on those that are most pertinent to our study.

Line 45: You haven't really established "dependency" of mangrove forest on sediment microbiome at this point. You can expand on how they can be considered dependent or remove this type of wording.
We agree with the reviewer and have added additional text and citations that establish the dependency of mangroves and microbes.

Line 47: I'm not sure what you mean by "single type of sediment." Since you don't discuss types of sediment or measure sediment characteristics such as grain size or grain type (silt, sand, etc.) in this paper you should not mention sediment type. I think you might mean that due to the fact that mangroves are highly influenced by tidal flow, which results in variations of "environmental conditions across small spatiotemporal scales,"
The reviewer is correct and we have removed the confusing text.

Line 60: What is the "original area"? Do you have specific information on this?
We have clarified the text to 'approximate pre-historical area' and have included the relevant citation.

Line 71: Not sure what you mean by "terrestrial processes." Do you mean biogeochemical processes?
We have corrected the text to refer to biogeochemical processes.

Line 74: I would say something more like "understand the differences between microhabitats within mangrove systems" instead.

We agree with the reviewer and we have reframed this to reflect the focus of previous work.

Line 75: By "mangrove regions" do you mean different tidal zones? If not, you might want to briefly explain what the different regions of mangroves are.

The reviewer is correct, we had intended this to be synonymous with 'tidal zones'. We have changed it to be 'tidal zones' to avoid confusion.

Line 78: Use something like "We identified taxa which may be driving different utrient cycles between zones." I should note that you may want to rethink this sentence altogether as you don't really show that there are different nutrient regimes/cycles between different zones as of now. You could include literature that suggests this or data of your own to support it.

We have rephrased this to emphasize that the taxa have different abundances between sites as well as making substantial inferred contributions to nutrient cycling.

Lines 81 -85: You could circle back to this in the discussion, specifically you could theorize what changes in community structure you might expect based on your findings and the literature in contaminated mangrove sediments.

We appreciate the reviewer's insight and have followed their suggestions in our revised Discussion section.

Methods

Line 100: What is meant by tidal influence? It would be good to include demarcations for this, i.e. distances, vegetation, etc. Based on Fig.1 it seems like you may have used sediment water content, or time of water coverage.

We agree with the reviewer that the estimation of tidal influence was incompletely explained. We have amended the text to include soil water content, coverage at time of collection, vegetative line and the agreement of a local guide.

Line 104: When you write "disruption of rhizospheres" do you mean "contamination" of rhizospheres associated with vegetation, because you aren't wanting to include those communities?

Yes, we wanted to avoid a possible interference of the rhizosphere microbiome on our analysis. We have clarified this in the text.

Line 107: Were vegetation densities measured, if so, what was the metric?

Vegetation densities were only qualitatively measured at the time of collection. We now state this in the text.

Line 111: How was organic matter content measured?

Organic matter was measured using the mass loss on ignition (LOI) method. We have amended the text to make this clear.

Line 127: I don't think you need the "ILLUMINACLIP" section, especially if you already have your code published on Github. Additionally, you explain your code in text immediately following. We agree with the reviewer and have removed this.

Line 131: You mention just QIIME, and QIIME2 in the following steps. QIIME is no longer supported or kept up, so if you used the original QIIME I would recommend using QIIME2 for that step.
We agree with the reviewer and have implemented the addition of QIIME2 for this step. Doing so we have seen an improvement in performance with an average of 2.1% more reads per sample than before. Correlation of abundances at the level of families resulted in a median $r^2 = 0.9995$ between the two methods.

Line 132 – 141: DADA2 calls ASVs and not OTUs. I have only superficially used QIIME2, as I typically use mothur or DADA2 directly in R, but my understanding is that QIIME2 and DADA2 primarily call ASVs, but gives the option to then cluster those ASVs after they have been called. If this is what you did, you should explain that process briefly. In reading on (Line 138), it would seem that there is either a miscommunication or misunderstanding of the bioinformatic steps with respect to clustering and taxonomic assignment. "Open reference" refers to a clustering method which can be done using Vsearch in QIIME2. It looks like when assigning taxonomy after OTU clustering, Q2 gives you 3 different option, I am thinking you used classify-consensus-vsearch? I would also combine sections 2.4.2 and 2.4.3 in to one section where all bioinformatics workflows in QIIME 2 are discussed together. For reference: https://docs.qiime2.org/2020.8/tutorials/otu-clustering/ and https://docs.qiime2.org/2020.8/tutorials/overview/

We apologize for the confusion. We had initially tested the performance of vsearch and the sklearn methods offered by QIIME2, and identified a degree of dissimilarity between results from the two methods, especially at the species level. Ultimately, we chose to use the sklearn approach based on the work of Bokulich et al. 2018 (https://doi.org/10.1186/s40168-018-0470-z). We have revised the text to reflect the exclusive use of the sklearn method.

Line 149 – 151: What dissimilarity metric did you use for metaMDS, i.e., jaccard, braycurtis etc.? I am wondering why you didn't run something like Canonical Analysis of Principle Coordinates (capscale in Vegan) to investigate correlations between environmental variables with community structure.

For metaMDS we used the Bray-Curtis metric. This is now stated in the methods section.

To address the reviewers concerns we also ran capscale . The model we started with was a basic linear module wherein all terms are additive and no terms are 'partialled out' (Oksanen, "Vegan: an introduction to ordination", https://cran.r-project.org/web/packages/vegan/vignettes/intro-vegan.pdf). Notably, we did not find this combined model to be statistically significant (Pr > 0.143) (figure below). Partialling out environmental variables resulted in increased model performance (Pr < 0.1 for Organic Matter

and Pr < 0.05 for Salinity). These agree with our own (unconstrained ordination) results using MetaMDS and Envfit.

All Variables
(Salinity + Organic_Matter + Temperature + Water_content)

|          | Df | SumOfSqs | F      | Pr(>F) |
|----------|----|----------|--------|--------|
| Model    | 2  | 1.3214   | 1.2026 | 0.143  |
| Residual | 6  | 3.2964   |        |        |

[Figure]

Salinity

|          | Df | SumOfSqs | F      | Pr(>F)  |
|----------|----|----------|--------|---------|
| Model    | 1  | 0.8307   | 1.5354 | 0.038 * |
| Residual | 7  | 3.7872   |        |         |

[Figure]

Organic Matter

|          | Df | SumOfSqs | F     | Pr(>F)  |
|----------|----|----------|-------|---------|
| Model    | 1  | 0.8185   | 1.508 | 0.087 . |
| Residual | 7  | 3.7994   |       |         |

[Figure]

Temperature

|          | Df | SumOfSqs | F    | Pr(>F) |
|----------|----|----------|------|--------|
| Model    | 1  | 0.6613   | 1.17 | 0.249  |
| Residual | 7  | 3.9565   |      |        |

[Figure]

Water Content

|          | Df | SumOfSqs | F      | Pr(>F) |
|----------|----|----------|--------|--------|
| Model    | 2  | 1.3214   | 1.2026 | 0.136  |
| Residual | 6  | 3.2964   |        |        |

[Figure]

Line 153: Did you take the limitations in the link provided below into consideration when running these analyses? PiCRUST2 Limitations Link:
http://picrust.github.io/picrust/tutorials/quality_control.html

We thank the reviewer for their suggestion and we have now taken these cautions into consideration. Because of the concerns surrounding under-represented taxa we did compare PICRUSt2 performance using the default NSTI cut-off of 2 and of 0.15. We found that the average Spearman *rho* correlation between the two sets was ~0.91, with a standard deviation of 0.01.

That said, the total ASVs retained using a 0.15 NSTI cutoff was only 12.5% - suggesting that the results (while similar by correlation) were not representative for the majority of taxa present in the sample. We have included this analysis in the Supplemental section.

However, it is important to note that, while the NSTI value of 0.15 is considered an upper bound at the taxonomic level of species (as per the PICRUSt1 manual), it is unclear what the acceptable upper bound would be for the taxonomic level of families, which is what we use in this paper. Indeed, while the median NSTI value for all ASVs in ~0.46 we find that the median minimum NSTI at the level of families is ~0.25.

Line 159: Explicitly explain how you conducted the taxa enrichment analysis. I could not find the reference paper for Spealman et. al. 2020.
We apologize to the reviewer for the confusion. Spealman et al. was a more expansive version of the supplemental information that was deposited in a citable archive upon submission of this paper as per Biogeosciences policy. We have now included this text in the supplement so that it may be more accessible to the reader and provided information on the taxa enrichment analysis in the Methods section.

**Results**

Line 186: Do you mean uncultured "prokaryote" not "eukaryote"? I don't believe there is any eukaryotic designations in the 16S SILVA taxonomy reference. You should also qualify further why you felt comfortable assigning an archaeal taxon to the uncultured "eukaryote".
Unfortunately, the current version of SILVA does have several mislabeled entries for 'eukaryote' under the Bathyarchaeia taxa (see below). These are certainly not eukaryotes but mislabeled entries, of which there are several in the Bathyarchaeia taxa. To prevent confusion we have changed the text such that label of "SILVA uncultured eukaryote (SUE)" will instead read "Uncultured Bathyarchaeia". We have included a short note as to the original mislabelling present in the Silva database.

[Figure]

Line 196: Figure 1B colors are difficult to differentiate. Consider adding a pattern to colors which are too close to differentiate. For figure 1B, consider changing the y-axis range to 32 so that we can see more of the other bars.

We agree with the reviewer that individual taxa were difficult to identify and we have corrected the figure.

Line 201, 204, 205: QIIME2 sentences belong in methods

We have moved sentences about QIIME2 to the methods section.

Line 207: P-value for Bray-Curtis is not significant

As per the reviewer comment for Line 211, we have removed this figure

Line 211: Why did you use both distance metrics, i.e., Jaccard and Bray-Curtis? You should choose the one most appropriate to your data and study. Did you try hierarchical clustering to see how data might cluster without apriori considerations like tidal zone?

We originally considered both the Bray-Curtis and Jaccard as they are different measures (quantitative and qualitative, respectfully). In the interests of concision we have removed the Bray-Curtis plot.

To address the reviewer's comment we performed hierarchical clustering (Gneiss, correlation clustering), but found that the balanced clusters only partially describe tidal zones (see below). This is not very surprising given that there is no statistically significant difference in their

[Figure]

pairwise combinations using the Beta-diversity tests.

Line 222: What is your organic matter (OM) metric, is it total OM? I don't think this is the best/clearest way to analyze these data with environmental variables. See methods comment Line 145-151.
As described above we used Loss on Ignition to measure organic matter content. We also evaluated these variables using Vegan's capscale method.

Line 227-229: These sentences belong in the methods section
We have revised the text by moving them to the methods section.

Line 231: What is "elevation" in this context? Do you mean zonation?
The reviewer is correct and we have revised the text to make this clear.

Line 236: What is an "icon" in this context?
We have now specified that the geometric shapes that are used are intended to show which pathways are enriched.

Line 237: What specifically about "carbon metabolism"? As all microbes need a carbon source, this should be explained in a bit more detail.
We agree with the reviewers and have revised our approach. Instead of looking at enrichment of KOs within broad functional categories we now look for enrichment within KEGG modules. This

has allowed us to describe the inferred functional enrichment in a more meaningful way. We have revised the figure, text, and methods to reflect this.

Line 247: In figure 5 it would be good to use different colors for differentiating bacteria and archaea as you are already using green and blue in the figure legend.
We agree with the reviewer that the color palette was confusing and have distinguished these domains in the figure.

Line 252: This sentence belongs in Methods
We have moved it to the methods.

Lines 255 & 256: don't need the "above both" phrase, it is confusing.
We agree with the reviewers and have removed this.

Line 261/262: "Taken together, KO enrichment reinforces the previously observed trend of reduced abundance in the Intertidal site, and greatest abundance at the Sublittoral zone." This sentence seems like it should be in the discussion, especially if the "previous trend" you are referring to is one form the literature. Also, be consistent on whether you are capitalizing the tidal zones or not. I think it is more correct not to capitalize.
We apologize for the confusion, we were referring to the observations of abundances we had described in an earlier section, not previous research. We have clarified the text.
We have revised the text throughout the manuscript to remove capitalization of the tidal zones.

Line 268-276: I may have missed the results in this paragraph, but it seems like all of this belongs in the methods section as it is describing how something was done versus reporting the findings of what was done.
We agree with the reviewer and have removed this paragraph.

Line 270: Where is here? Is it this study or a figure?
This line has been removed.

Lines 277 – 327: I would combine all of the "cycling" sections under one section called "biogeochemical cycling" or something like this. You have several sentences throughout this section that would be more appropriate in the discussion section. Essentially any of the sentences with citations should probably be in the discussion. Examples: Lines 287, 291-293, etc.
We agree with the reviewer and have condensed this section. As the citations were intended to provide additional lines of evidence to support the potential metabolic functional analysis we have moved the most relevant ones to the discussion and included the rest as supplemental information.

Line 328: I appreciate the effort that went into this figure. Personally, I would like to see this figure with less taxa, only ones you specifically mention within the text, so that it is less busy. I am not sure why the # of nodes legend needs to be a tapered triangle. It makes me think I

should be considering both the color and thickness of lines when I'm looking at nodes. I also think you should use a more differentiating way to denote taxa with an associated metabolic role in the literature and taxa with KO greater than 10%. You could use black and white circles, and add a gray circle for those with both if they exist.

We agree that the figure is overly complex and have followed the reviewer's suggestions in making a simpler version with cleaner layout and reduced text.

**Discussion**

I decided to make an overarching comment here, instead of going line by line, because the discussion needs a lot of work and restructuring. I think one of the best ways to go about fixing the discussion will be to come up with a thesis statement for each paragraph and figure out what points you are trying to convey. This will help you to clarify and re-organize your thoughts. I would like to see in the discussion more synthesis of your findings, such as why you think you might find certain taxa with potential metabolic capabilities enriched in certain tidal zones. It seems to me that you set out to study the sediment microbiome of pristine mangrove environment across 3 tidal zones to serve as a baseline and to characterize differences in taxa and potential biogeochemical cycling that is predominant in those zones. However, neither your introduction or discussion provide enough clear, relevant background or support for your overarching goal. I would have also liked to see some discussion on anaerobic taxa, and where you find more anaerobic taxa with respect to tidal zone. You us the term microhabitat in both the intro and discussion, but it is unclear what this means in the context of your study. You should clearly define what your usage of microhabitat means. Are you talking about microbial habitat, are you taking about spatial scales, millimeters – meters?

In accordance with the reviewer's comments, the discussion section has been restructured in order to provide a better synthesis of our findings with respect to the hypothesis and prior literature. We have also revised the introduction to provide the relevant background for the overarching goals of the study and a definition of the concept of microhabitats that we are using. We believe that these substantive changes have addressed the reviewer's concern and have significantly enhanced the clarity of our manuscript.

**Technical Corrections:**

Line 57: Latter not "later"
We have corrected the text.

Line 62: I am assuming that the A in APA is referring to Ambiente, but just want to point out you use the English "Environmental" just before APA, so I'm not sure if you should write EPA or use the word Ambiente.
We agree with the reviewer that the original Portuguese is 'Área de Proteção Ambiental' - however, we have tried to faithfully translate this into English as 'Environmental Protection Area' while retaining the un-Anglicized acronym. Similar to how Germany retains the DEU abbreviation. We will amend the text with the full phrase in Portuguese.

Line 117: Use protocol instead of "program"

We have changed the wording to "protocol".

Line 154: All KEGGs should be capitalized. Line 193: Genera instead of "genus".

The wording has been corrected throughout the text.

Line 101: I don't know if superficial is the correct word, I typically see the use of "surficial".

The text has been corrected.

---

## Referee Report (RR1)

There are a few areas where tense usage is not consistent.

Line 25 : "characterize" should be "characterized"
Line 26 : " find" should be "found"

Lines 37-38: I think it should be "Mangroves span tidal zones that are characterized by periodic tidal flooding, such that environmental conditions, such as salinity, vary greatly across small spatiotemporal scales.

Line 39: This sentence is a bit awkward because it sounds like you are describing and impact. Instead, you could say something about what tidal zones mangroves span. This might be good as you don't really mention Sublittoral, Intertidal, or Supralittoral anywhere in your introduction.

Lines 54-56: Awkward wording, try to make a little clearer.

Line 69: "as" should be "is"

Line 79: Remove "Importantly"

Line 108: make sure you put "gene" after "16S rRNA"

Line 129: Make sure you mention that vegan is an R package. You don't seem to mention R in this section or the citation for R. Additionally Jari Oksanen is the creator of the vegan package. I'm not sure where the Dixon 2003 citation comes from.
https://www.rdocumentation.org/packages/vegan/versions/2.4-2

Line 135 & 136: "As we are using 16S rRNA amplicon sequencing a crucial limitation must be considered in evaluating our results". The tense is incorrect and it would be good to elaborate a little more on what the crucial limitation is and why it should be considered.

Lines 196 – 200: It seems like you need to report more information on these results. What are the 8 families you found to be significantly different between/amongst zones? What 4 families were least abundant in the intertidal zone?

Lines 201-206: Needs clarification similar to comment above. Which 7 families contributed to carbon metabolism, etc. It is also important to cite the figure at some point in this paragraph.

Line 207: Spacing between words is off

Lines 299-300. The last sentence is a bit repetitive and unclear

Figure 3. It is a good idea to include the number of dimensions used and the stress for the nMDS analysis

Figure 4. When you say "To have been labelled with a metabolic pathway" are you referring to pathway enrichment? If so maybe point that out so that the reader can see the connection between this and the figure legend.

---

## Editor Decision (ED1)

[revised manuscript text omitted]

**Micrococcaceae***
Pentose phosphate (M00006)

**Flavobacteriaceae**
Pyruvate metabolism (M00168, M00579)
Sulfur metabolism (M00176)
Pentose phosphate (M00580)

**Clostridiaceae 1***
Citrate cycle (TCA cycle) (M00307)

**Rhodobacteraceae***
Sulfur metabolism (M00595)

**Desulfobacteraceae***
Amino acid metabolism (M00020)
Nitrogen metabolism (M00175)
Citrate cycle (TCA cycle) (M00307)
Sulfur metabolism (M00596)

**Vibrionaceae**
ATP synthesis (M00150)

**Bacillaceae**
Citrate cycle (TCA cycle) (M00003, M00011)
Amino acid metabolism (M00021)
Cofactor and vitamin metabolism (M00140)
ATP synthesis (M00157)
Pyruvate metabolism (M00168, M00579)
Sulfur metabolism (M00176, M00616)
Nitrogen metabolism (M00531, M00615)
Pentose phosphate (M00004, M00005, M00006, M00007, M00345, M00580)

Sublittoral | Intertidal | Supralittoral

○ Carbohydrate metabolism    ▲ Carbon fixation    ● Carbon metabolism
◉ Nitrogen metabolism    ◉ Sulfur metabolism    ⬡ Other metabolism

Fig. 5. Zone specific enrichment of pathway modules. Here, we show the impact that the differential abundances of major families (Figure 4) have on potential functional profiles at each zone at the level of pathway modules. To be included here each module must be completely enriched within a single taxon in a single zone, such that the taxa accounts for at least 10% of the total potential functional abundance for each KO involved in the pathway. Families with a median NSTI within 1 standard deviation of 0.15 are labelled with *.

[Figure]

**Fig. 6. Zone specific measures of metabolism associated KEGG Orthologs.** The heatmaps show the metabolism associated KOs that have significant differences of functional abundance between zones for carbon, phosphorus, sulfur and nitrogen pathways. Notably, the majority of KOs (45/54, 83%), have their highest relative functional abundance in the sublittoral zone, with the intertidal (5/54, 9.3%) and supralittoral (4/54 7.4%) as near equal minorities.

590

---

## Author Response (AR2)

We thank the reviewer for their time and review of our paper. We have made the revisions as per their suggestions and feel the manuscript is improved as a result.

There are a few areas where tense usage is not consistent.
We have revised the text and corrected the tense usage.

Line 25 : "characterize" should be "characterized"
The text has been corrected.

Line 26 : " find" should be "found"
The text has been corrected.

Lines 37-38: I think it should be "Mangroves span tidal zones that are characterized by periodic tidal flooding, such that environmental conditions, such as salinity, vary greatly across small spatiotemporal scales.
We agree with the reviewer and have revised the sentence.

Line 39: This sentence is a bit awkward because it sounds like you are describing and impact. Instead, you could say something about what tidal zones mangroves span. This might be good as you don't really mention Sublittoral, Intertidal, or Supralittoral anywhere in your introduction.
We agree with the reviewer and have revised the text to better define the tidal zones studied. We have also included them in the abstract to help the reader.

Lines 54-56: Awkward wording, try to make a little clearer.
We have revised the sentence.

Line 69: "as" should be "is"
The text has been corrected.

Line 79: Remove "Importantly"
We have removed the word from the text.

Line 108: make sure you put "gene" after "16S rRNA"
We revised and corrected the text.

Line 129: Make sure you mention that vegan is an R package. You don't seem to mention R in this section or the citation for R. Additionally Jari Oksanen is the creator of the vegan package. I'm not sure where the Dixon 2003 citation comes from.
https://www.rdocumentation.org/packages/vegan/versions/2.4-2

We apologize for the oversight and have amended the text and included citations to R (and python). While the Dixon citation is the initial announcement of the vegan package, we agree that it is not uptodate and have also included a contemporary citation.

Line 135 & 136: "As we are using 16S rRNA amplicon sequencing a crucial limitation must be considered in evaluating our results". The tense is incorrect and it would be good to elaborate a little more on what the crucial limitation is and why it should be considered.
We have revised the sentence and elaborated the text in order to make it clearer for the readers.

Lines 196 – 200: It seems like you need to report more information on these results. What are the 8 families you found to be significantly different between/amongst zones? What 4 families were least abundant in the intertidal zone?
We have revised the text and added more details to the reported results in the section.

Lines 201-206: Needs clarification similar to comment above. Which 7 families contributed to carbon metabolism, etc. It is also important to cite the figure at some point in this paragraph.
We have added more details to the reported results and a citation to the corresponding figure.

Line 207: Spacing between words is off
We have corrected the formating.

Lines 299-300. The last sentence is a bit repetitive and unclear
We agree with the reviewer and have revised the text to make it clearer.

Figure 3. It is a good idea to include the number of dimensions used and the stress for the nMDS analysis
We agree with the reviewer and have updated the figure, adding the dimensions parameter and stress values.

Figure 4. When you say "To have been labelled with a metabolic pathway" are you referring to pathway enrichment? If so maybe point that out so that the reader can see the connection between this and the figure legend.
We agree with the reviewer that the text was unclear, we have revised the caption to read; "Pathway enrichment requires taxa to have at least 10% of three KOs in that pathway in any zone."

**Anonymous Referee #3**

**Review:** Biogeosciences "Tidal influences on the structure and function of prokaryotic communities in the sediments of a pristine Brazilian mangrove"

The authors present a study that is highlighting differences in community composition in the sediment of mangroves across different tidal zones. Apart from focusing on the community composition the authors also try to predict community functioning by predicting functions from 16S rRNA gene sequences via Picrust2. In the updated version of the manuscript the limitations of predicting functions based on taxonomic affiliation are better addressed than in the original manuscript. However, some critical points of the manuscript need to be revised and addressed.

We thank the reviewer for their time and insightful commentary. The paper has gained scope and breadth based on their feedback and we hope that these improvements show.

**Comments**

Abstract and Introduction

- The abstract would benefit from some sharpening and highlighting actual findings, such as abundant taxa discovered and how that differs from other studies in mangroves that discovered much greater diversity in mangrove sediments (see e.g. Zhou et al., 2017, Zhang et al. 2021. We agree with the reviewer and have revised the abstract to reflect more the main findings and how they compare with similar studies. A portion of this comparison is with the Zhou data and is included in our Supplemental as Figures 11, 12, and 13. However, we urge caution in the comparison, as no sublittoral measure was made and we can only infer that the mud flat described by Zhou can be treated as an intertidal equivalent. Further comparisons (although not re-analysis of data) with more or less similar datasets were also made for numerous other studies as is reflected in Supplemental File 2.

- The introduction seems to have improved significantly compare to the original version of the manuscript. However, it could still need some sharpening. Especially the explanation of study site, aim and hypotheses could be brought more "to the point" to become clearer to the reader. We appreciate the reviewers' kind words as to our improvement. We have revised the text in order to make the central points clearer to the readers.

Results
- Figure 2: I find this way of plotting the abundance data very hard to read compared to a normal stacked bar chart. E.g. finding the abundances of Bathyarchaeia is almost impossible. For the main text a simple stacked bar chart on phylum level might be easier to read and interpret. The figure has been changed and is now displaying the taxonomic levels of phylum and class. The figure with taxonomic families has been moved to the supplemental material.

- under 3.3. the enrichment of specific taxa and their potential function across zones is highlighted. This paragraph would benefit greatly if the connections between tidal zone,

dominant taxa and function would become more clear and easier to understand. At this point the text is hard to read with numerous pathway codes and repetitions.

We apologize for the difficulty. We have revised the figure to help make the context more clear and have edited the text to remove unnecessary pathway codes.

Figure 5 does support the reader in understanding the text, however also here highlighting the different predicted functions more clearly could be helpful.

We agree with the reviewer and have revised the text and figure to avoid repetitions and excessive citation of codes. The figure 5 has been modified to allow a better understanding of the results described in the text.

- What about archaeal taxa found in the sediment?

Unfortunately, we did not find any archaea that satisfied the statistical conditions to be included in this step of the analysis. This may in part be due to the incomplete representation of them in our 16S rRNA gene amplicon library, as the reviewer points out. To aid the reader in this, we have added clarifying text about the archaeal taxa to the results and discussion sections.

**Discussion**

- How does the community composition, especially in the sublittoral zone, compare to other environments such as shallow marine sediments in coastal areas?

We have added select papers from coastal marine studies to our literature review and have added to our discussion how these distinctly different environments compare to mangrove sediments, according to the literature.

- Studies that have performed metagenome sequencing on mangrove sediments are more suitable to predict functions. How to the findings of this study compare to those of other studies (e.g. Zhang et al., 2021). Trying for highlight differences and similarities would certainly benefit the manuscript.

We agree with the reviewer and have revised the text in order to add comparisons between our results and other work that sought to predict microbial functions in the sediments, using both metagenomic and 16S rRNA gene amplicons. We do, indeed, find a significant difference in our samples being predominantly *Firmicute* instead of *Proteobacteria,* an interesting difference.

- Recent studies using metagenomics on mangroves, but also other environments have shown, that especially yet uncultivated microorganisms are often being not detected by primer sets frequently used for amplicon sequencing (Eloe-Fadrosh et al., 2016). However, these microbes are often dominant parts of the microbial community. Could key players in mangrove sediments have been overlooked?

We agree with the reviewer and have emphasized the important limitations of the use of 16S rRNA gene amplicon sequencing for the identification of some relevant taxa in these microbiomes.

**Minor comments:**

- the correct term is "16S rRNA gene amplicon sequencing"
We have corrected the term throughout the manuscript.

- Supplemental table 1: change "metagenomics method" to just "Method". DGGE, T-RFLP and amplicon sequencing are not metagenomics! More recent metagenomic studies are completely missing from this table as well as being ignored in the manuscript.
We have changed the term and revised the table to add more recent microbiome studies.

- Figure 1: stay consistent between capital and lower-case letters in figure and figure legend (e.g. (a) vs (A))
We have revised the text and corrected inconsistencies.

---

## Author Response (AR3)

Dear Dr Gresham and co-authors.

Thank you for revising your paper based on the reviewers' comments. You have done a great job at addressing their concerns and the paper is greatly improved by these revisions. Therefore, I am pleased to accept your paper for publication in Biogeosciences!!

I did find a few places that would benefit from technical corrections as indicated in the annotated PDF attached. In some cases these comments might not be ideal so I tried to indicate as such by saying "optional". Please let me know if you have any questions.

Also, I think the inclusion of so many supplemental files is really helpful for scientific transparency and allowing others to more easily compare their results/data to yours. Nice job with that.

Best,

Dr. Denise Akob

Dr. Denise Akob,

Thank you for working with us on this paper, we appreciate the time and energy you have spent improving it, finding reviewers, and keeping the project moving forward. The paper has dramatically improved because of your, and the reviewers, support.

We have made all of the corrections you had advised.

Thank you again